# Development of equation of motion deciphering locomotion including omega turns of *Caenorhabditis elegans*

**Taegon Chung, Iksoo Chang, Sangyeol Kim***

Daegu Gyeongbuk Institute of Science and Technology, Daegu, Republic of Korea

**Abstract** Locomotion is a fundamental behavior of *Caenorhabditis elegans* (*C. elegans*). Previous works on kinetic simulations of animals helped researchers understand the physical mechanisms of locomotion and the muscle-controlling principles of neuronal circuits as an actuator part. It has yet to be understood how *C. elegans* utilizes the frictional forces caused by the tension of its muscles to perform sequenced locomotive behaviors. Here, we present a two-dimensional rigid body chain model for the locomotion of *C. elegans* by developing Newtonian equations of motion for each body segment of *C. elegans*. Having accounted for friction-coefficients of the surrounding environment, elastic constants of *C. elegans*, and its kymogram from experiments, our kinetic model (ElegansBot) reproduced various locomotion of *C. elegans* such as, but not limited to, forward-backward-(omega turn)-forward locomotion constituting escaping behavior and delta-turn navigation. Additionally, ElegansBot precisely quantified the forces acting on each body segment of *C. elegans* to allow investigation of the force distribution. This model will facilitate our understanding of the detailed mechanism of various locomotive behaviors at any given friction-coefficients of the surrounding environment. Furthermore, as the model ensures the performance of realistic behavior, it can be used to research actuator-controller interaction between muscles and neuronal circuits.

*For correspondence:
sykim@dgist.ac.kr

**Competing interest:** The authors declare that no competing interests exist.

## eLife assessment

This **useful** study introduces a simple mechanical model of *C. elegans* locomotion that captures aspects of the worm's behavioral repertoire beyond forward crawling. While the kinetic model (ElegansBot) provides a compromise and starting point to help understand the mechanical components of *C. elegans* behavior, the claim that this work improves on extant mechanical models is **incomplete**, including modeling a 3-dimensional turning behavior with a 2-dimensional model without sufficient justification. In addition, the results of the application of the model to previously unstudied behaviors are primarily qualitative and do not produce new predictions.

## Introduction

With only a few hundred neurons, *Caenorhabditis Elegans* (*C. elegans*) perform various behaviors such as locomotion, sleeping, reproduction, and hunting (**Hall and Altun, 2008**). The connectome structure among 302 neurons and 165 somatic cells of *C. elegans* was discovered by pioneering works (**Cook et al., 2019**; **White et al., 1986**). *C. elegans* is a cost-efficient and widely used model in neuronal research. Its small body size and minimal nutritional requirements contribute to its cost efficiency. The organism matures in a shorter period, about three days, compared to other model animals such as fruit flies or mice. Its transparent body allows for easy microscopic observation of its internal structures or artificially expressed green fluorescent proteins. Moreover, due to the hermaphroditic nature of *C.*

*elegans*, offspring mostly share the same genotype as the parent, which simplifies the multiplication of the worm population for research purposes (*Hall and Altun, 2008*).

*C. elegans* bends its body with a sinusoidal wave pattern when moving forward or backward. The driving force for this movement comes from the difference between perpendicular and parallel frictional forces, which it experienced from a surrounding environment. This thrust force pushes the worm along the ground surface with which the worm contacts (*Berri et al., 2009*; *Boyle et al., 2012*; *Hu et al., 2009*; *Niebur and Erdös, 1991*). Even if a worm has a sinusoidal modulation generated inside it, it has difficulties in forward and backward locomotion if it does not feel the difference in frictional forces from its surroundings.

Mechanical simulators of rod-shaped animals such as *C. elegans* (*Boyle et al., 2012*; *Niebur and Erdös, 1991*), fish (*Ekeberg, 1993*), and snakes *Hu et al., 2009* have been used in various studies. These simulators demonstrate how the activities of muscle cells are represented as behavioral phenotypes, which are determined by signals from a neuronal circuit simulator (*Boyle et al., 2012*; *Ekeberg, 1993*; *Niebur and Erdös, 1991*). They also show how muscle cells return proprioceptive signals back to the neuronal circuit simulator and how animals intentionally distribute body weight for locomotion patterns (*Hu et al., 2009*). Similarly, the kinematic simulator of fish (*Ekeberg, 1993*), which has a locomotion pattern in that the animal mostly undulates in a particular direction, was used with a neuronal network simulator to model the undulation of swimming behavior. This combination of kinematic simulator and neuronal network simulator was also used to model how the locomotion pattern changes due to a surrounding environment (*Boyle et al., 2012*) and how the central pattern generator arises from a few cells (*Boyle et al., 2012*; *Izquierdo and Beer, 2018*).

Even though there were studies on kinematic simulation of rod-shaped animals (*Boyle et al., 2012*; *Ekeberg, 1993*; *Hu et al., 2009*), to our best knowledge, there was no kinetic model that reproduces complex locomotion behavior of *C. elegans,* which includes all of the various modes of locomotion of *C. elegans* such as forward locomotion, backward locomotion, and turn from experimental observations. Instead, muscle cell activities from Ansatz (*Hu et al., 2009*), a hypothesis of the solution, or signals from a neuronal circuit simulator (*Boyle et al., 2012*; *Ekeberg, 1993*) were applied to the kinematic simulators. A simulator should have an operational structure that imitates physical quantities from an experiment to reproduce the motion of *C. elegans* in the experiment. However, until now, no kinetic simulation has such a structure. If there is a simulator that reproduces the motion of individual

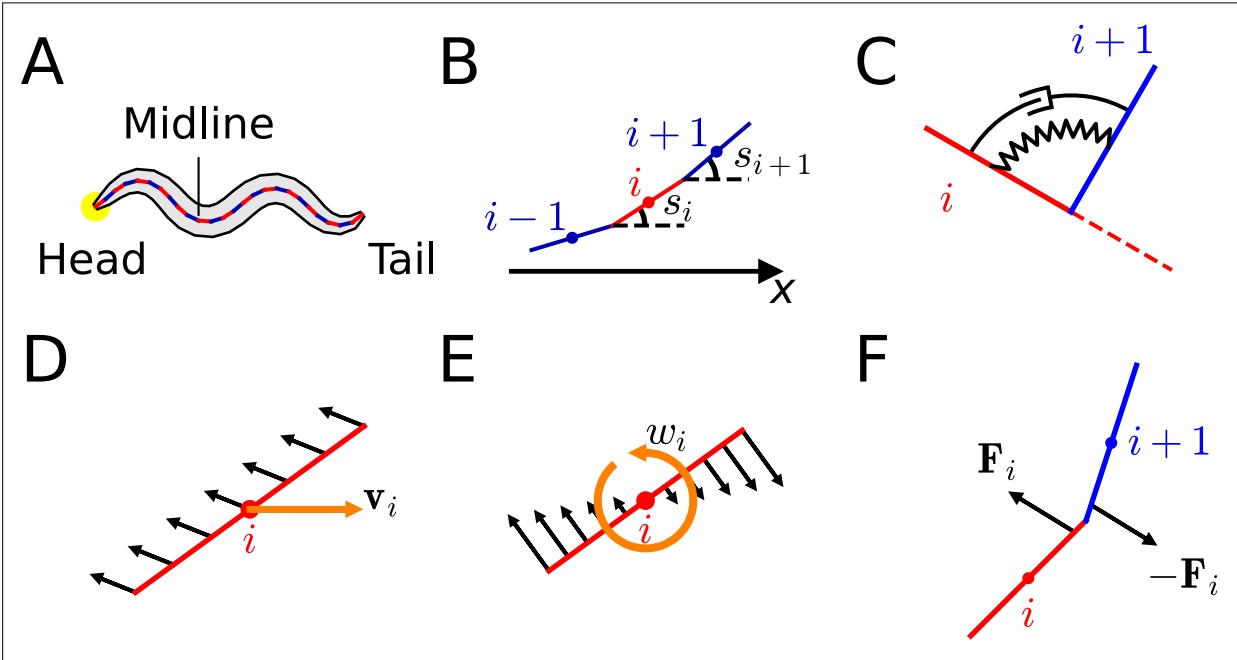

**Figure 1.** Components of ElegansBot. (**A**) Chain model for *C. elegans* body. (**B**) Rods in chain model. (**C**) i-actuator, which is a damped torsional spring. (**D**) Frictional force (black arrow) due to the translation motion of a rod. (**E**) Frictional force (black arrow) due to the rotational motion of a rod. (**F**) Joint force (black arrows) acting on i-rod and (i+1)-rod.

experiments, analysis of the kinetics of motion of specific experiments, which provides information on the individual force that exerts on each body part of the animal, will be enabled. Also, as the kinetic simulation reproduces the motion of *C. elegans*, the behavioral phenotype that emerged from the muscle activity of neuronal circuit simulation will be more credible.

We built a Newtonian-mechanics two-dimensional rigid body chain model of *C. elegans* to reproduce its locomotion. We incorporated its body angle, related to the contraction of the body wall muscle of *C. elegans*, into the primary operating principle of our kinetic model so that the model simulates measurable physical quantities of *C. elegans* from its experimental video. The model includes a chain of multiple rod rigid bodies, a damped torsional spring between the rigid bodies, and a control angle, which is the dynamic baseline angle from the value of the kymogram of a physical experiment. We formulated Newtonian equations of translational and rotational motion of the rigid body model and computed the numerical solution of the equations by numerical integration using the semi-implicit Euler method. As a result, we were able to demonstrate trajectories and kinetics of the general locomotion of *C. elegans*, such as crawling, swimming (**Vidal-Gadea et al., 2011**), omega-turn, and delta-turn (**Broekmans et al., 2016**).

## Results

### Newton's equation of motion for locomotion of *Caenorhabditis elegans:* How does ElegansBot work?

We introduce the simple chain model of *C. elegans'* body. *C. elegans* has an elongated body along the head-to-tail axis. Thus, the worm's body can be approximated as a midline extended along the anterial-posterial axis in the xy-coordinate plane (**Figure 1A**). Let $M$ (=2 µg, details in 'Worm's mass, actuator elasticity coefficient, and damping coefficient' of Appendix) be the mass and $L$ (=1 mm) be the length of the worm. Midline was approximated as $n$ (=25) straight rods, whose ends are connected to the ends of neighboring rods (**Figure 1A**). The mass, length, and moment of inertia of each rod is $m = M/n$ , $2r = L/n$ , and $I = mr^2/3$, respectively. When numbering the rods in order, with the rod at the end of the head being labeled as '1-rod' and the rod at the end of the tail being labeled as 'n-rod,' let us designate the i-th rod as 'i-rod.' The point where i-rod and (i+1)-rod meets is 'i-joint.'.

The motion of the worm corresponds to the motion of all the rods. To describe the motion of each rod (i-rod), we need to determine the displacement vector ($\mathbf{d}_i$), velocity vector ($\mathbf{v}_i$), the angle measured counterclockwise from the positive x-axis to the tangential direction of the rod ($s_i$) (**Figure 1B**), and angular velocity ($\omega_i$) of i-rod at a given time $t$. However, the minimum information required to describe the motion of all rods includes the displacement vector ($\mathbf{d}_c$) and velocity vector ($\mathbf{v}_c$) of the worm's center of mass, $s_i$ and $\omega_i$ for each rod (Details in 'Minimum information required to describe the motion of each rod' of Appendix).

Value of time-dependent variables such as $\mathbf{d}_c$ , $\mathbf{v}_c$ , $s_i$ , and $\omega_i$ at a given time, $t$ will be expressed as $*^{(t)}$. When initial values, $\mathbf{d}_c^{(0)}$ , $\mathbf{v}_c^{(0)}$ , $s_i^{(0)}$ , and $\omega_i^{(0)}$ are given, the Newtonian equation of motion for acceleration, $\mathbf{a}_c$ and angular acceleration, $\{\alpha_i\}_{i \in 1, \cdots, n}$ must be acquired and numerically integrated twice to find $\mathbf{d}_c^{(t)}$ , $\mathbf{v}_c^{(t)}$ , $s_i^{(t)}$ , and $\omega_i^{(t)}$ at a given time, $t$. To obtain the Newtonian equations of motion, we must find every force and torque acting on each rod. There are frictional force, muscle force, and joint force among types of forces acting on the rod, and there are frictional torque, muscle torque, and joint torque among types of torques whose descriptions are as follows.

The only external force acting on the worm is a frictional force from a ground surface such as an agar plate or water. The frictional force is an anisotropic Stokes frictional force, with a magnitude proportional to the speed and assumed different friction coefficients in perpendicular and parallel directions (**Boyle et al., 2012**), which guarantees that linearity in velocity is preserved in frictional force as well (Details in 'Preservation of linearity in friction' of Appendix). Because of this preservation of linearity, the frictional forces of translational motion (**Figure 1D**) and rotational motion (**Figure 1E**) can be calculated separately and added together to find total frictional force and torque. Previously known values of the friction coefficients in perpendicular and parallel directions are used (**Boyle et al., 2012**).

Let the perpendicular and parallel friction coefficients be $b_\perp$ and $b_\parallel$ for a straightened worm, respectively. Each rod experiences $1/n$ of the frictional force the worm gets. Thus, the perpendicular and parallel friction coefficients of each rod are $b_\perp/n$ , $b_\parallel/n$ , respectively. The ratio of perpendicular

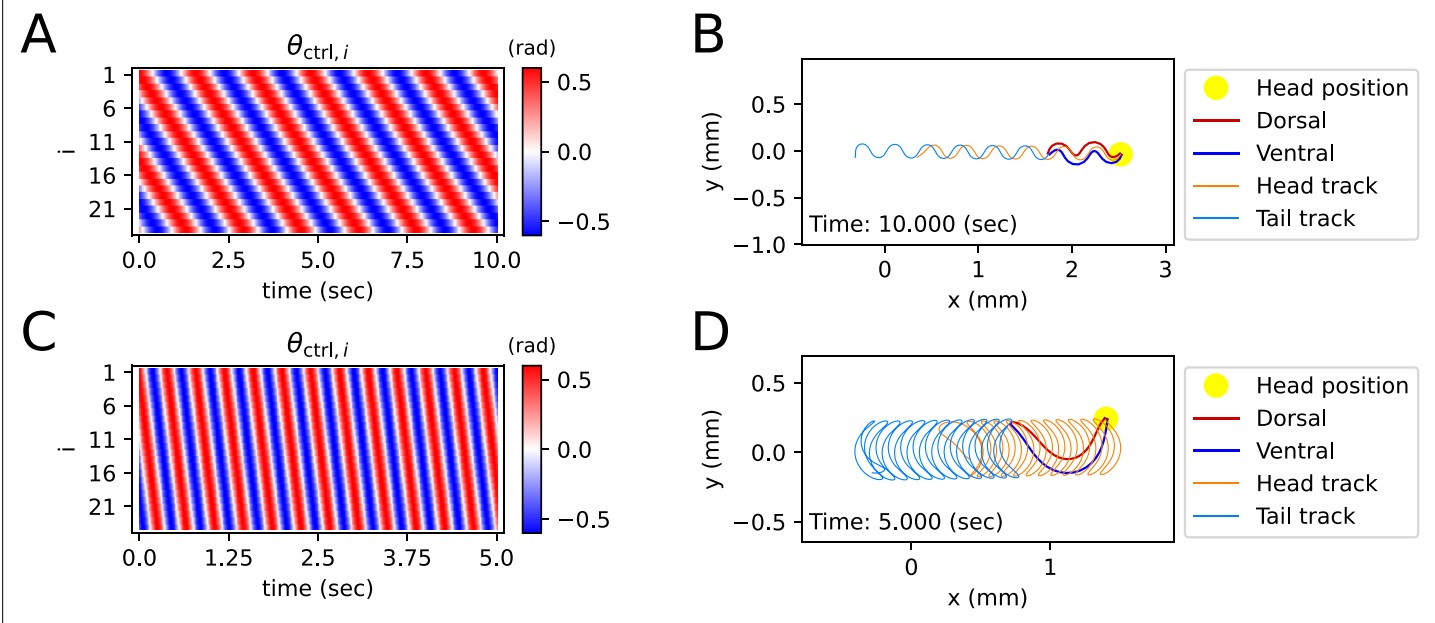

**Figure 2.** Simulated locomotion from a sine kymogram. (**A**) Crawling kymogram. Kymogram indicates the angle of i-joint which is located between i-rod and (i+1)-rod. Red and blue color mean i-joint bend in the dorsal and ventral directions, respectively. (**B**) Crawling trajectory. The yellow circle indicates the position of the worm's head. The Orange and sky-blue lines show the worm's head and tail trajectories, respectively. (**C**) Swimming kymogram. (**D**) Swimming trajectory.

The online version of this article includes the following figure supplement(s) for figure 2:

**Figure supplement 1.** Locomotion propulsion mechanism.

friction coefficient to parallel friction coefficient ($b_\perp/b_\parallel$) is 40 in agar plate and 1.5 in water (***Berri et al., 2009***; ***Boyle et al., 2012***). This ratio is an important determining factor in whether the locomotion would be crawling or swimming (***Boyle et al., 2012***). The total frictional force that i-rod receives is $\mathbf{F}_{b,i} = -\frac{b_\parallel}{n} \left( \mathbf{v}_i \cdot \widehat{\mathbf{r}}_i \right) \widehat{\mathbf{r}}_i - \frac{b_\perp}{n} \left( \mathbf{v}_i \cdot \widehat{\mathbf{N}}_i \right) \widehat{\mathbf{N}}_i$ ('·': dot product of vectors, $\widehat{\mathbf{r}}_i \equiv \begin{bmatrix} \cos(s_i) & \sin(s_i) \end{bmatrix}^\mathsf{T}$: unit vector parallel to i-rod, $\widehat{\mathbf{N}}_i \equiv \begin{bmatrix} -\sin(s_i) & \cos(s_i) \end{bmatrix}^\mathsf{T}$: unit vector perpendicular to i-rod), and the total frictional torque that i-rod receives is $\tau_{b,i} = -\frac{1}{3} \frac{b_\perp}{n} r^2 \omega_i$ (positive or negative values are for torque pointing away from or into the paper plane, respectively.) (The proof is in 'Frictional torque by rotational motion' of Appendix).

Mature hermaphrodite *C. elegans* has four muscle strands at the left dorsal, right dorsal, left ventral, and right ventral part of the body, and each muscle strand has 24, 24, 23, and 24 muscle cells, respectively (***White et al., 1986***). Muscle cells at similar positions on the anterior-posterior axis have an activity pattern in that muscles on one side (either dorsal or ventral) cooperate, and those on the opposite side have alternative activities. (***Hall and Altun, 2008***).

Therefore, we modeled a group of about four muscle cells, which are left dorsal, right dorsal, left ventral, and right ventral, at the same position on the anterior-posterior axis as one actuator (***Figure 1C***) so that there is a total of 24 ($\simeq (24 + 24 + 23 + 24)/4$) actuators in the worm. On i-joint of the chain, there is an actuator labeled as i-actuator. As the number of actuators is 24, we set the number of rod($n$) as 25, which is one more than the number of actuators. The actuator was modeled as a damped torsional spring due to the viscoelastic characteristics of muscle (***Boyle et al., 2012***; ***Hill, 1938***). If the dorsal muscles of i-actuator contract more than the ventral muscles, i-actuator will bend to the dorsal direction and vice versa. To express this phenomenon by an equation, we defined the torque that i-actuator exerts on i-rod as $\tau_i = \tau_{\kappa,i} + \tau_{c,i}$ that the elastic term is $\tau_{\kappa,i} = \kappa \left( \theta_i - \theta_{\mathrm{ctrl},i} \right)$ and the damping term is $\tau_{c,i} = c \left( \omega_{i+1} - \omega_i \right)$ where $\theta_{\mathrm{ctrl},i}$ is control angle, $\theta_i = s_{i+1} - s_i$, and $\kappa$ and $c$ are the elasticity and damping coefficients of an actuator, respectively.

Control angle ($\theta_{\mathrm{ctrl},i}$) is a variable inside the elastic part of the muscle torque ($\tau_{\kappa,i}$), to which $\tau_{\kappa,i}$ drives $\theta_i$ close. Also, the control angle ($\theta_{\mathrm{ctrl},i}$), which can be expressed by a heatmap (***Figures 2A, C,***

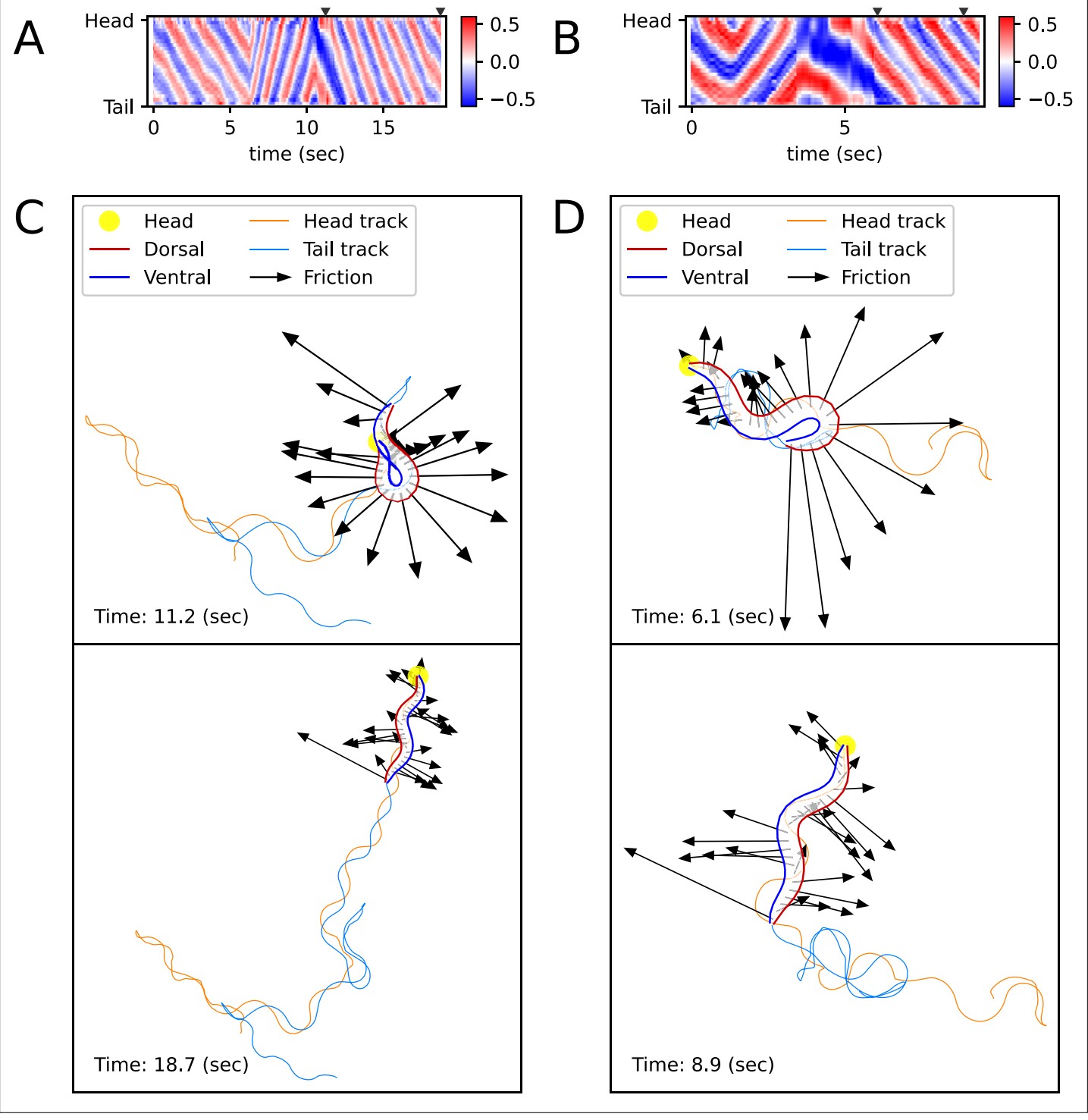

**Figure 3.** Simulated locomotion from a kymogram of a real worm locomotion video. The length and direction of a black arrow indicate the magnitude and direction of the frictional force ($-\mathbf{F}_{b,i}^{(t)}$) that the corresponding body part, which is the starting point of the arrow, exerts on the surface. (**A**) Escaping behavior kymogram. Triangles over the heatmap indicate the corresponding time of snapshots shown in Figure (**C**). (**B**) Delta-turn kymogram. Triangles over the heatmap indicate the corresponding time of snapshots shown in Figure (**D**). (**C**) Escaping behavior trajectory. (**D**) Delta-turn trajectory. The arrow length scale is different from Figure (**C**) to clearly show the arrows' directions and head and tail tracks.

The online version of this article includes the following video(s) for figure 3:

**Figure 3—video 1.** Reproduced escaping behavior of experimental video (*Broekmans et al., 2016*).

https://elifesciences.org/articles/92562/figures#fig3video1

*Figure 3 continued on next page*

*Figure 3 continued*

**Figure 3—video 2.** Reproduced delta-turn of experimental video (*Broekmans et al., 2016*).

https://elifesciences.org/articles/92562/figures#fig3video2

3A and B), is an input value based on experimental data, a numerical model, or a neuronal network model. $\tau_{c,i}$ represents the damping effect of muscle cells and somatic cells near i-actuator. The elasticity coefficient ($\kappa$) and damping coefficient ($c$) of an actuator were induced from previously known values (*Boyle et al., 2012*) (Details in 'Worm's mass, actuator elasticity coefficient, and damping coefficient' of Appendix).

By assuming that i-rod receives torque ($\tau_i$) from i-actuator, and (i+1)-rod receives torque ($-\tau_i$), we can depict the bending that arises from the differential contraction of the dorsal and ventral muscles in i-actuator. The total muscle torque that i-rod receives from damped torsional springs on both ends is $\tau_{c\kappa,i} = \tau_i - \tau_{i-1}$. The total muscle force ($\mathbf{F}_{c\kappa,i}$) that i-rod receives from both of its ends is as follows (Details in 'Proof of muscle force' of Appendix).

$$\mathbf{F}_{c\kappa,i} = \sum_{j=-1}^{0} \frac{(-1)^{j-1} \tau_{i+j} \sin \frac{\theta_{i+j}}{2}}{r \cos^2 \frac{\theta_{i+j}}{2}} \begin{bmatrix} \cos \left( \frac{s_{i+j} + s_{i+j+1}}{2} \right) \\ \sin \left( \frac{s_{i+j} + s_{i+j+1}}{2} \right) \end{bmatrix}$$

Two neighboring rods (i-rod and (i+1)-rod) are connected at i-joint. Therefore, when a force is applied to i-rod, (i+1)-rod also receives distributed force (*Figure 1F*) which we name as 'joint force.' The joint force that (i+1)-rod exerts on i-rod is symbolized as $\mathbf{F}_i \equiv \mathbf{F}_{(i+1)i}$. By Newton's third law of motion about action and reaction, the joint force that i-rod exerts on (i+1)-rod is $\mathbf{F}_{i(i+1)} = -\mathbf{F}_{(i+1)i} = -\mathbf{F}_i$ (*Figure 1F*). Joint force ($\mathbf{F}_i$) can be calculated from the previously introduced given values ($s_i$, $\mathbf{F}_{c\kappa,i}$, $\mathbf{F}_{b,i}$, $\tau_{c\kappa,i}$, $\tau_{b,i}$) (Details in 'Joint force calculation method' of Appendix). When $\mathbf{F}_0 = \mathbf{F}_n = \mathbf{0}$, then the total joint force that i-rod receives is $\mathbf{F}_{\text{joint},i} = \mathbf{F}_i - \mathbf{F}_{i-1}$ and the total torque caused by joint force is $\tau_{\text{joint},i} = \left[ \mathbf{r}_i \times \left( \mathbf{F}_i + \mathbf{F}_{i+1} \right) \right]$ where '×' between two vectors means cross-product.

As all forces and torques are found, $\mathbf{d}_c^{(t)}$, $\mathbf{v}_c^{(t)}$, $s_i^{(t)}$, $\omega_i^{(t)}$ can be calculated by solving translational and rotational Newtonian equations of motion with numerical integration. The time-step ($\Delta t$) used in this work is $10^{-5}$ s unless otherwise noted. Because the only external force exerts on the worm is the frictional force, the equation of translational motion is $\mathbf{a}_c = \frac{\sum_i \mathbf{F}_{b,i}}{M}$. If friction coefficients ($b_\perp$, $b_\parallel$) are significantly greater than $\frac{M}{\Delta t}$, numerical integration using the explicit Euler method ($\mathbf{v}_c^{(t+\Delta t)} = \mathbf{v}_c^{(t)} + \mathbf{a}_c^{(t)} \Delta t = \mathbf{v}_c^{(t)} + \frac{\sum_i \mathbf{F}_{b,i}}{M} \Delta t$) becomes unstable (*Butcher, 2003*). So, we tackled this instability of numerical integration by developing semi-implicit Euler Method ($\mathbf{v}_c^{(t+\Delta t)} = \mathbf{v}_c^{(t)} + \mathbf{a}_c^{(t+\Delta t)} \Delta t \simeq \mathbf{v}_c^{(t)} + \frac{1}{1+\frac{b_\perp \Delta t}{M}} \frac{\sum_i \mathbf{F}_{b,i}^{(t)}}{M} \Delta t$), which makes numerical integration stable when any frictional coefficients greater than or equal to 0 is given (Details in 'Proof of numerical integration for the translational motion of a worm using semi-implicit Euler method' of Appendix).

The equation of rotational motion of i-rod is $I\alpha_i = \tau_{\text{total},i} = \tau_{b,i} + \tau_{c\kappa,i} + \tau_{\text{joint},i}$. When the friction-related value ($b_\parallel r^2$), elasticity-related value ($\kappa \Delta t$), or damping coefficient($c$) is significantly larger than $\frac{I}{\Delta t}$, numerical integration using explicit Euler method ($\omega_i^{(t+\Delta t)} = \omega_i^{(t)} + \alpha_i^{(t)} \Delta t = \omega_i^{(t)} + \frac{\tau_{\text{total},i}^{(t)}}{I} \Delta t$) becomes unstable (*Butcher, 2003*). To solve this instability, we constructed a semi-implicit Euler method for rotational motion and an error-corrected equation for angular momentum (Details in 'Numerical integration of the rotational motion of i-rod using semi-implicit Euler method' and 'Correction formula for the rotational inertia of the entire worm' of Appendix). By using these semi-implicit Euler methods, solutions for $\mathbf{d}_c^{(t)}$, $\mathbf{v}_c^{(t)}$, $s_i^{(t)}$, $\omega_i^{(t)}$ of a worm at a given time can be available for the ground surface of agar whose $b_\perp$, $b_\parallel$ are significantly larger than $\frac{M}{\Delta t}$, water which has smaller friction coefficients than agar, or frictionless ground surface.

## Can *C. elegans* in ElegansBot crawl or swim?

A kymogram is a heatmap that shows body angle, $\theta_i^{(t)}$ ($i \in \{1, \cdots, n-1\}$) at a given time, $t$. By fitting a sine function to the kymogram of previous work (*Vidal-Gadea et al., 2011*), we obtained linear-wavenumber (after this referred to as wavenumber) and period of *C. elegans* crawling on the agar plate and swimming in water. The wavenumber ($\nu$) and the period ($T$) are, respectively, 1.832 and 1.6 (s) on

the agar plate and 0.667 and 0.4 (s) in water. For both crawling and swimming, amplitude ($A$) was set to 0.6 (rad) arbitrarily to match the trajectory shown in the experimental video (*Vidal-Gadea et al., 2011*). Each kymogram of crawling (*Figure 2A*) and swimming (*Figure 2C*) was calculated by substituting amplitude ($A$), wavenumber ($\nu$), and period ($T$) into into $\theta_{\text{ctrl},i}^{(t)} = A \cos\left(2\pi\left(\nu\left(i-1\right)/\left(n-2\right) - t/T\right)\right)$.

Crawling trajectory, which performs sinusoidal locomotion in the positive x-axis direction, was obtained by inputting a crawling kymogram as $\theta_{\text{ctrl},i}^{(t)}$ input to ElegansBot (*Figure 2B*). Regarding crawling, the head track and the tail track have similar shapes. However, the tail track is more toward the negative x-axis direction than the head track. The difference between the head and tail tracks indicates that the worm pushes the ground surface by the distance between the head track and tail track to obtain thrust (*Figure 2B*). Indeed, we found that the body part placed diagonally with respect to the direction of the worm's locomotion is pushing along the ground surface (*Figure 2—figure supplement 1A*). The thrust force of the worm cancels out most of the drag force, which enables the worm to move at nearly constant velocity. The average velocity of the worm is 0.208 (mm/s), which is consistent with the known values (*Cohen et al., 2012*; *Jung et al., 2016*; *Omura et al., 2012*; *Shen et al., 2012*).

In the previous work, the worm showed swimming behavior in a water droplet on an agar plate (*Vidal-Gadea et al., 2011*). As the friction coefficient of water is smaller than that of agar, even though the area that the worm swept was wider during swimming than crawling, the worm did not move forward much in comparison to the area it swept (*Figure 2D*). The worm gained significant momentum in the forward direction of locomotion when the body bent in the c-shape (*Figure 2—figure supplement 1B*). In contrast to crawling, during swimming, the worm did not receive constant thrust force over time. Thus, the speed of the worm exhibited significant oscillations over time (*Figure 2—figure supplement 1B*), and the average velocity was 0.223 (mm/s).

## ElegansBot exhibits more complex behavior including the turn motion

Unlike previous *C. elegans* body kinematic simulation studies, our simulation can replicate the worm's behavior using a kymogram (*Figure 3A and B*) derived from experimental videos. We utilized open-source software, Tierpsy Tracker (*Javer et al., 2018*) and WormPose (*Hebert et al., 2021*), to obtain the kymogram input ($\theta_{\text{ctrl},i}^{(t)}$) for the ElegansBot. Through simulation, we aimed to reproduce the omega-turn and delta-turn behaviors observed in the experimental videos (*Broekmans et al., 2016*). When we used the vertical and horizontal friction coefficients $b_\perp$ and $b_\parallel$ on agar, as proposed in the previous work (*Boyle et al., 2012*), the trajectory was not accurately replicated. Given that the friction coefficients could vary depending on the concentration of the agar gel, we used $b_\perp/100$ and $b_\parallel/100$ for the vertical and horizontal friction coefficients, respectively, which resulted in a better trajectory replication (Details in 'Proper selection of friction coefficients' in Appendix).

The trajectory (*Figure 3C and D*, *Figure 3—videos 1 and 2*) obtained from ElegansBot accurately reproduces the experimental video (*Broekmans et al., 2016*). The changes in the direction of movement caused by turns are well replicated. Additionally, during the omega-turn or delta-turn, the body briefly performs a deep bend, and we newly discovered the mechanism that gains significant propulsion from the deep bend region to change direction using ElegansBot (*Figure 3C and D*). Moreover, the ElegansBot accurately reproduces not only the turns but also complex behaviors like the sequence of forward-backward-turn-forward, also known as escaping behavior.

Additionally, we calculated the mechanical power of the worm as a quantitative indicator to explain its locomotion during sequenced locomotive behavior, based on behavior classification (forward, backward locomotion, or turn, as defined in Methods). During escaping behavior, the worm produced an average power of 2094 fW in the initial forward locomotion, followed by an average of 16,437 fW (7.85 times that of the initial forward locomotion) in backward locomotion, and an average of 11,118 fW (5.31 times that of the initial forward locomotion) during turning (*Figure 4A*). After turning and resuming forward locomotion, it produced an average power of 5480 fW (2.62 times that of the initial forward locomotion). This indicates that the worm produced more power than that of initial forward locomotion to escape sudden threats. Let's denote the average of a quantity for all given $i$ as $\langle * \rangle_i$. At the moment the worm formed a deep bend ($t=11.2$ s), the average magnitude of frictional force of the body part forming the deep bend ($\left\langle \left| \mathbf{F}_{b,i}^{(t)} \right| \right\rangle_i$ where $i$=4 to 15) was 3536 pN, compared to the average magnitude of the remaining parts ($\left\langle \left| \mathbf{F}_{b,i}^{(t)} \right| \right\rangle_i$ = 1737 pN where $i$=1 to 3 or $i$=16 to 25), which was 2.04 times greater (*Figure 3C*, *Figure 4A*). We analyzed delta-turn in the same manner. The worm

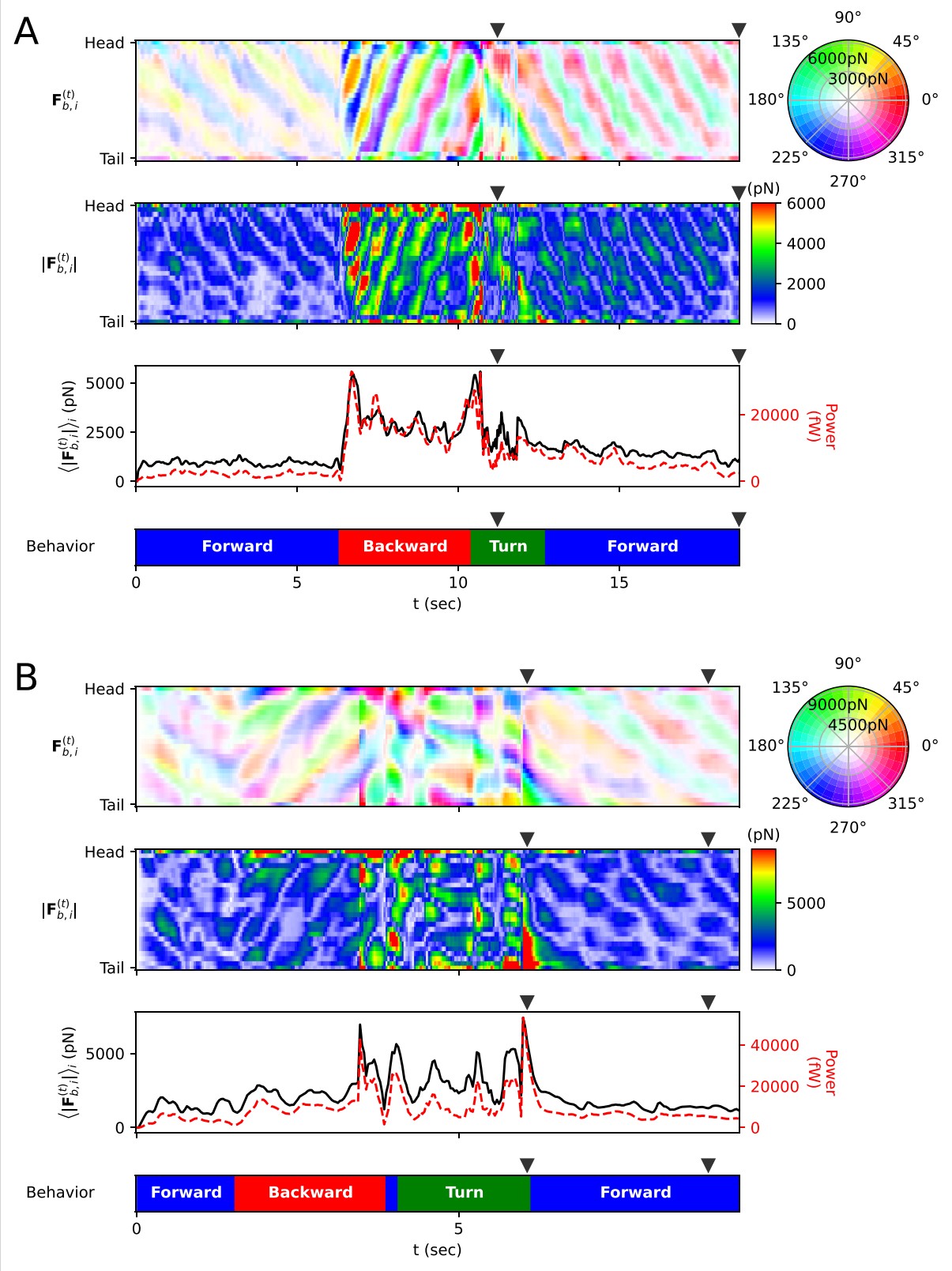

**Figure 4.** Frictional force on each rod. (**A**) Escaping behavior. The top panel represents the frictional force $\mathbf{F}_{b,i}^{(t)}$ experienced by i-rod. As indicated on the color wheel to the right, the hue of this heatmap represents the direction of the force, and the saturation represents the magnitude of the force. The second panel from the top shows the magnitude of the frictional force $\left|\mathbf{F}_{b,i}^{(t)}\right|$. The third panel from the top represents the average $\left\langle\left|\mathbf{F}_{b,i}^{(t)}\right|\right\rangle_i$ (black

*Figure 4 continued on next page*

*Figure 4 continued*

solid line) of each column in the middle panel and the power (red dotted line), which is the amount of energy the worm consumes per unit time. The bottom panel represents the classification of the worm's behavior (blue: forward locomotion, red: backward locomotion, green: turn) (The definitions of behavioral categories are in Methods). The triangles over each panel indicate the corresponding time of the snapshots depicted in *Figure 3C*. (**B**) Delta-turn. The triangles over each panel indicate the corresponding time of the snapshots depicted in *Figure 3D*.

The online version of this article includes the following figure supplement(s) for figure 4:

**Figure supplement 1.** Process of defining behavioral categories.

produced an average power of 3514 fW in the initial forward locomotion, followed by an average of 11,176 fW (3.18 times that of the initial forward locomotion) in subsequent backward locomotion. In the relatively short duration of forward locomotion following the backward locomotion, the worm produced an average power of 17,544 fW (4.99 times that of the initial forward locomotion), and an average of 13,046 fW (3.71 times that of the initial forward locomotion) during turns (*Figure 4B*). After the turn, when resuming forward locomotion, the worm produced an average power of 6429 fW (1.83 times that of the initial forward locomotion). At the moment the worm formed a deep bend (t=6.1 s), the average magnitude of frictional force of the body part ($\left\langle \left| \mathbf{F}_{b,i}^{(t)} \right| \right\rangle_i$ where i=16 to 25) was 10,497 pN, compared to the average magnitude of the remaining parts ($\left\langle \left| \mathbf{F}_{b,i}^{(t)} \right| \right\rangle_i$ = 2,677 pN where i=1 to 15), which was 3.92 times greater (*Figure 3D*, *Figure 4B*). In both escaping behavior and delta-turn, the worm consistently produced more power in the subsequent backward locomotion and turn than in the initial forward locomotion.

## ElegansBot presents body shape ensembles of *C. elegans* from a shape in water en route to agar

While there have been studies on how locomotion patterns change in agar and water by merging neural and kinematic simulations (*Boyle et al., 2012*), there have been none that solely used kinetic simulation to analyze how speed manifests depending on the frequency and period of locomotion. We demonstrate this aspect. We studied the locomotion speed of the worm under different friction coefficients, which represent the influence of water, agar, and intermediate frictional environment, using ElegansBot. The vertical and horizontal friction coefficients in water are $b_{\text{water},\perp} = 5.2 \times 10^3$ (µg/sec) and $b_{\text{water},\|} = b_{\text{water},\perp}/1.5$, respectively, while in agar, these values are $b_{\text{agar},\perp} = 1.28 \times 10^8$ (µg/sec) and $b_{\text{agar},\|} = b_{\text{agar},\perp}/40$ (*Boyle et al., 2012*). For environmental index $\sigma \in [0, 1]$, we have defined the vertical and horizontal friction coefficients in the environment between water ($\sigma = 0$) and agar ($\sigma = 1$) as $b_{\sigma,\perp} = b_{\text{water},\perp}^{1-\sigma} b_{\text{agar},\perp}^{\sigma}$ and $b_{\sigma,\|} = b_{\text{water},\|}^{1-\sigma} b_{\text{agar},\|}^{\sigma}$, respectively.

Under an environmental index $\sigma$, for various pairs of frequency-period ($\nu$, $T$) when the control angle is $\theta_{\text{ctrl},i}^{(t)} = A \cos \left( 2\pi \left( \nu \frac{i-1}{23} - \frac{t}{T} \right) \right)$ (with $A = 0.6$ (rad)), we have found the ($\nu$, $T$) that maximizes the worm's average velocity(optimal ($\nu$, $T$)) (*Figure 5A*). The optimal ($\nu$, $T$) exhibits a nearly linear distribution (*Figure 5B*). We noticed a transition from swimming body shape to crawling body shape as $\sigma$ varies (*Figure 5*, *Figure 5—figure supplement 1*). The optimal ($\nu$, $T$) for $\sigma$=0(water) is (0.65, 0.4 s), matching the actual ($\nu$, $T$) value of swimming behavior (*Vidal-Gadea et al., 2011*). The optimal ($\nu$, $T$) for $\sigma$=1(agar) is (1.9, 0.8 s), and the optimal $\nu$ (1.9) matches the actual $\nu$ value (1.832) for crawling behavior (*Vidal-Gadea et al., 2011*), with the optimal $T$ (0.8 s) being half the actual $T$ value (1.6 s).

We wanted to understand the impact of the environmental index $\sigma$ not only on forward locomotion but also on sequenced locomotive behavior. First, we analyzed the effect of the environmental index $\sigma$ on escaping behavior as follows. Let's denote the set of a quantity for all pairs of index $i$ and time $t$ as $\{*\}_{i,t}$. When the escaping behavior kymogram input $\left\{ \theta_{\text{ctrl},i}^{(t)} \right\}_{i,t}$ was same as *Figure 3A*, we explored the effect of vertical and horizontal friction coefficients on the worm's motion. Where $\sigma$ ranged from 1.0 to 0, the trajectory varied with $\sigma$ (*Figure 5—figure supplement 2A*), and $E_\theta = \left\langle \left| \theta_i^{(t)} - \theta_{\text{ctrl},i}^{(t)} \right| \right\rangle_{i,t}$ decreased as $\sigma$ decreased (*Figure 5—figure supplement 2B*). From $\sigma = 1.0$ to $\sigma = 0.1$, the total absolute angular change ($S = \sum_{t=0}^{T-\Delta t} \left| \left\langle s_i^{(t+\Delta t)} \right\rangle_i - \left\langle s_i^{(t)} \right\rangle_i \right|$ where $T$ is the total time of the experimental video.) increased as $\sigma$ decreased. However, from $\sigma = 0.7$ to $\sigma = 0$, $S$ remained constant within the error of 0.33 rad, and the total traveled distance($\sum_t \left| \mathbf{v}_c^{(t)} \Delta t \right|$) decreased as $\sigma$ decreased. The maximum total traveled distance was at $\sigma = 0.8$. Using the same analysis method with the kymogram

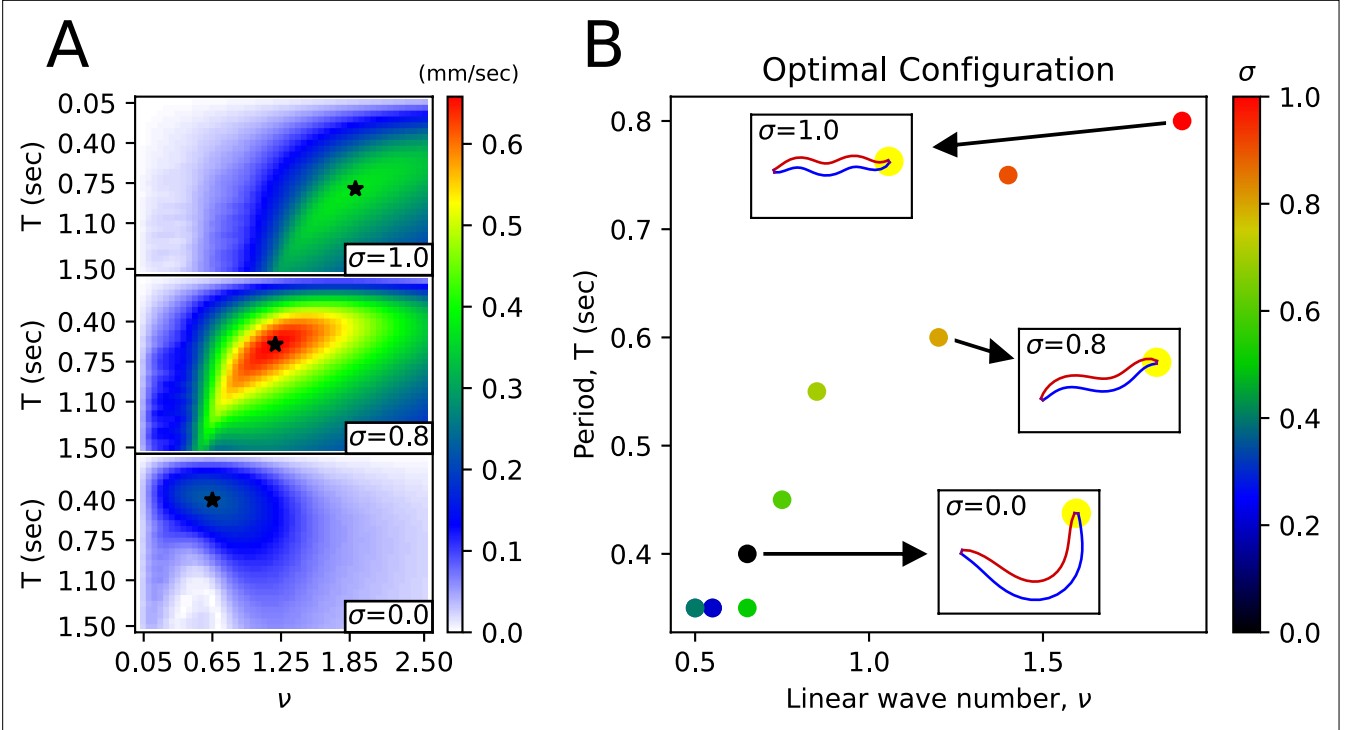

**Figure 5.** Body shape transition from the shape in water to the shape in agar. (**A**) Average velocity of the worm as a function of wavenumber ($\nu$) and period ($T$) for a given friction coefficient. The star symbol indicates the pair of ($\nu$, $T$) that maximizes the worm's average velocity. (**B**) For each environmental index $\sigma$, the pair of ($\nu$, $T$) that maximizes the worm's average velocity. The worm figures inside the small rectangles pointed to by the arrows represent the body shape corresponding to the respective ($\nu$, $T$) pair.

The online version of this article includes the following figure supplement(s) for figure 5:

**Figure supplement 1.** Transition of body shape from water ($\sigma = 0$) to agar ($\sigma = 1$).

**Figure supplement 2.** The effect of the environmental index $\sigma$ on escaping behavior.

**Figure supplement 3.** The effect of the environmental index $\sigma$ on the delta-turn.

input $\left\{ \theta_{\mathrm{ctrl},i}^{(t)} \right\}_{i,t}$ same as *Figure 3B*, we analyzed the impact of the environmental index $\sigma$ on delta-turn. Where $\sigma$ ranged from 1.0 to 0, the trajectory varied with $\sigma$ (*Figure 5—figure supplement 3A*), and $E_\theta$ also decreased as $\sigma$ decreased (*Figure 5—figure supplement 3B*). From $\sigma = 1$ to $\sigma = 0.6$, $\mathcal{S}$ increased as $\sigma$ decreased. From $\sigma = 0.6$ to $\sigma = 0$, $\mathcal{S}$ decreased as $\sigma$ decreased. The maximum total traveled distance was at $\sigma = 0.9$. From $\sigma = 0.9$ to $\sigma = 0$, the total traveled distance decreased as $\sigma$ decreased.

## Discussion

### ElegansBot is an advanced kinetic simulator that reproduces *C. elegans*' various locomotion

The known crawling speed range of *C. elegans* (*Cohen et al., 2012*; *Jung et al., 2016*; *Omura et al., 2012*; *Shen et al., 2012*) matches the speed in our simulation. The force dispersion pattern of the forward movement of a snake (*Hu et al., 2009*) is similar to the force dispersion pattern of crawling in our model, where the body part placed diagonally in the direction of movement generates thrust. The head and tail tracks of our simulation resemble the trace left on the agar plate by *C. elegans* during locomotion (*Yeon et al., 2018*), providing evidence of the mechanism where *C. elegans* moves forward by pushing along the ground surface. Given that friction and elasticity coefficients can vary between experiments, the appropriate selection of these values allows the trajectories of omega-turns and delta-turns in our simulations to match the experimental videos (*Broekmans et al., 2016*). Previous work (*Berri et al., 2009*; *Boyle et al., 2012*) eliminated inertia from the equations of motion, but our simulation includes it, allowing calculation even in cases where inertia is significant due to low

friction coefficients. Using the crawling and swimming wavenumbers and periods from the experiments (*Vidal-Gadea et al., 2011*), we computed sine functions to create trajectories for crawling and swimming. We also analyzed how friction forces act on the worm during crawling and swimming, studying how the worm gains propulsion. We demonstrated that we could reproduce various locomotion observed in experimental videos, such as forward-backward-(omega turn)-forward constituting escaping behavior and delta-turn navigation, by providing the kymogram obtained from representative physical values from the experimental videos, as well as the kymogram obtained from a program (*Hebert et al., 2021*; *Javer et al., 2018*) extracting the body angles from actual experimental videos into ElegansBot. Our established Newtonian equations of motion are accurate and robust, suggesting that not only does our simulation replicate the experimental videos, but it also provides credible estimates for detailed forces.

## ElegansBot will serve as a strong bridge for enhancing the knowledge in 'from-synapse-to-behavior' research

Our method could be used for kinetic analysis of behaviors not covered in this paper. It could also be used when analyzing behavior changes caused by mutation or ablation experiments. Given that our simulation allows for kinetic analysis, it could be used to calculate the energy expended by the worm during locomotion, serving as an activity index. Our simulation only requires the body angles as input data, so even if the video angle shifts and trajectory information is lost, the trajectory can be recovered from the kymogram. Our simulation could also be used when studying neural circuit models of *C. elegans*. It could be used to check how signals from neural network models manifest as behaviors which is a needed function from previous work (*Sakamoto et al., 2021*), and it could be used when studying compound models of neural circuits and bodies. For example, when creating models that receive proprioception input based on body shape (*Boyle et al., 2012*; *Ekeberg, 1993*; *Izquierdo and Beer, 2018*; *Niebur and Erdös, 1991*), our method could be used. Finally, our method could be used in general for the broad utility to analyze the motion of rod-shaped animals like snakes or eels and to simulate the motion of rod-shaped robots.

## Methods

### Frequency and wavelength of *C. elegans* locomotion

Sine function fitting was applied to the crawling and swimming kymograms (*Vidal-Gadea et al., 2011*) to determine the frequency and wavelength of *C. elegans* locomotion on agar and water.

### *C. elegans* locomotion videos

Videos of *C. elegans*' escaping behavior and foraging behavior were obtained from previous work (*Broekmans et al., 2016*). A single representative video out of a total of one hundred escaping behavior videos was used as data in this paper. Additionally, only the delta-turning portion of the foraging behavior videos was cut out and used as data in this paper.

### Obtaining kymograms from video

The following method was used to extract the kymogram from the video of *C. elegans*: The body angles and midline were extracted using Tierpsy Tracker (*Javer et al., 2018*) from the original video where the worm is locomoting. Tierpsy Tracker failed to extract the midline of the worm when the body parts meet or the worm is coiled. The midline information of the frames successfully predicted by Tierpsy Tracker and the original video information were used as ground truth training data for a program called WormPose (*Hebert et al., 2021*). WormPose trained an artificial neural network to extract the body angles and midline of a coiled worm using a generative method based on the input data. The body angles were extracted from the original video using the trained WormPose program. For the frames where Tierpsy Tracker failed to extract the body angles, it was replaced with the body angles extracted by WormPose. Nonetheless, there were frames where the body angle extraction failed. If the period of failed body angle prediction was continuously less than three frames (about 0.01 s), the body angles for that period was predicted using linear interpolation.

### Program code and programming libraries

Equations for the chain model, friction model, muscle model, and numerical integration that constitute the ElegansBot were designed from the body angle information and kymogram that change

every moment of time. Python (*van Aken et al., 1995*) version 3.8 was used to implement the equations constituting ElegansBot as a program. NumPy (*Harris et al., 2020*) version 1.19 was used for numerical calculations, and Numba (*Lam et al., 2015*) version 0.54 was used for CPU calculation acceleration. SciPy (*Virtanen et al., 2020*) version 1.5 was used for curve fitting and Savitzky-Golay filter (*Savitzky and Golay, 1964*) to classify the worm's behavioral categories. The Matplotlib (*Hunter, 2007*) library was used to represent *C. elegans*' body pose and trajectory in figures and videos. The program code used in the research can be obtained from the open database GitHub (Taegon Chung, 2023, ElegansBot, 1.0.1, https://github.com/taegonchung/elegansbot, copy archived at *Chung, 2024*) or Python software repository PyPI (https://pypi.org/project/ElegansBot/), and the web live demo can be found at GitHub Page (https://taegonchung.github.io/elegansbot/). This code calculated the ElegansBot simulation of 10 s of simulation time in about 10 s of run-time on an Intel E3-1230v5 CPU.

## Physical constants of the ground surface

The friction coefficient values for the ground surface where *C. elegans* crawled and swam and the elastic and damping coefficients of *C. elegans* muscles were obtained from previous work (*Boyle et al., 2012*). The muscle elasticity and damping coefficients were converted into coefficients for the damped torsional spring to be used in our model (Details in 'Worm's mass, actuator elasticity coefficient, and damping coefficient' of Appendix).

## Defining behavioral categories

We determined the classification of the worm's behavior over time as follows. Let $\xi_i \equiv (i-1)/(n-2)$ (i.e., $0 \leq \xi_i \leq 1$). We defined a sine function fitting for the body angle $\theta_i^{(t)}$ as $\hat{\theta}_i^{(t)} = A^{(t)} \sin\left(\left(2\pi/\lambda^{(t)}\right)\left(\xi_i - \xi_0^{(t)}\right)\right) + \theta_0^{(t)}$ (where $A^{(t)} \geq 0$ and $0 \leq \xi_0^{(t)} < \lambda^{(t)}$). Let us denote the set of a quantity for all $i$ as $\{*\}_i$. For a given time $t$, by curve fitting the function $\hat{\theta}_i^{(t)}$ to the set of body angles $\left\{\theta_i^{(t)}\right\}_i$, we can obtain the parameters ($A^{(t)}$, $\lambda^{(t)}$, $\xi_0^{(t)}$, $\theta_0^{(t)}$) (*Figure 4—figure supplement 1*). For curve fitting, we used 'curve_fit' from the scipy library (*Virtanen et al., 2020*). 'curve_fit' requires the function to be fitted, the data, and the initial guess values of the function parameters. Therefore, we determined the initial guess values ($\hat{A}^{(t)}$, $\hat{\lambda}^{(t)}$, $\hat{\xi}_0^{(t)}$, $\hat{\theta}_0^{(t)}$) as follows. For $t = 0$, we calculated ($\hat{A}^{(0)}$, $\hat{\lambda}^{(0)}$, $\hat{\xi}_0^{(0)}$, $\hat{\theta}_0^{(0)}$) using the following equations:

$$\hat{A}^{(0)} = \frac{1}{2}\left(\max_i(\theta_i^{(0)}) - \min_i(\theta_i^{(0)})\right)$$

$$\hat{\lambda}^{(0)} = \frac{2}{n-2}\left(\operatorname*{argmax}_i(\theta_i^{(0)}) - \operatorname*{argmin}_i(\theta_i^{(0)})\right)$$

$$\hat{\xi}_0^{(0)} = \frac{1}{n-2}\left(\operatorname*{argmax}_i(\theta_i^{(0)})\right) - \frac{\hat{\lambda}^{(0)}}{4}$$

$$\hat{\theta}_0^{(0)} = \frac{1}{n-1}\left(\sum_i \theta_i^{(0)}\right)$$

Using these initial guess values, we curve-fitted $\hat{\theta}_i^{(0)}$ to $\{\theta_i^{(0)}\}_i$ to obtain ($A^{(0)}$, $\lambda^{(0)}$, $\xi_0^{(0)}$, $\theta_0^{(0)}$). For $t \geq \Delta t$, we obtained the initial guess values for curve fitting as $\hat{A}^{(t)} = A^{(t-\Delta t)}$, $\hat{\lambda}^{(t)} = \lambda^{(t-\Delta t)}$, $\hat{\xi}_0^{(t)} = \xi_0^{(t-\Delta t)}$, $\hat{\theta}_0^{(t)} = \theta_0^{(t-\Delta t)}$. Then, using these initial guess values, we curve-fitted $\hat{\theta}_i^{(t)}$ to $\left\{\theta_i^{(t)}\right\}_i$ to obtain ($A^{(t)}$, $\lambda^{(t)}$, $\xi_0^{(t)}$, $\theta_0^{(t)}$). Since the phase $\xi_0^{(t)}$ is not continuous for all time $t$, we defined a continuous value $\tilde{\xi}_0^{(t)}$ for all time $t$ as follows (*Figure 4—figure supplement 1*):

$$\tilde{\xi}_0^{(t)} = \begin{cases} \xi_0^{(t)} & \text{if } t = 0 \\ \xi_0^{(t)} - \xi_0^{(t-\Delta t)} + \tilde{\xi}_0^{(t-\Delta t)} & \text{if } t > 0 \text{ and } -\frac{\lambda^{(t)}}{2} \leq \xi_0^{(t)} - \xi_0^{(t-\Delta t)} \leq \frac{\lambda^{(t)}}{2} \\ \xi_0^{(t)} - \xi_0^{(t-\Delta t)} + \tilde{\xi}_0^{(t-\Delta t)} + \lambda^{(t)} & \text{if } t > 0 \text{ and } \xi_0^{(t)} - \xi_0^{(t-\Delta t)} < -\frac{\lambda^{(t)}}{2} \\ \xi_0^{(t)} - \xi_0^{(t-\Delta t)} + \tilde{\xi}_0^{(t-\Delta t)} - \lambda^{(t)} & \text{if } t > 0 \text{ and } \frac{\lambda^{(t)}}{2} < \xi_0^{(t)} - \xi_0^{(t-\Delta t)} \end{cases}$$

To obtain the derivatives of the noise-reduced smoothed values $\bar{\xi}_0^{(t)}$ and $\bar{\theta}_0^{(t)}$ for the raw data $\tilde{\xi}_0^{(t)}$ and $\theta_0^{(t)}$, respectively, we applied a Savitzky-Golay filter (*Savitzky and Golay, 1964*; *Virtanen et al., 2020*). This filter, set with a smoothing time window of 0.5 s, a polynomial order of 2, and a derivative

order of 1, yielded $\frac{\mathrm{d}\bar{\xi}_0^{(t)}}{\mathrm{d}t}$ and $\frac{\mathrm{d}\bar{\theta}_0^{(t)}}{\mathrm{d}t}$ . We then calculated the temporal integrals $\xi'_0{}^{(\zeta)} \equiv \sum_{t=0}^{\zeta} \frac{\mathrm{d}\bar{\xi}_0^{(t)}}{\mathrm{d}t} \Delta t$ and $\theta'_0{}^{(\zeta)} \equiv \sum_{t=0}^{\zeta} \frac{\mathrm{d}\bar{\theta}_0^{(t)}}{\mathrm{d}t} \Delta t$. Let us denote the average of a quantity for all time $t$ as $\langle * \rangle_t$. We calculated $\bar{\xi}_0^{(t)} = \xi'_0{}^{(t)} - \left\langle \xi'_0{}^{(t)} \right\rangle_t + \left\langle \tilde{\xi}_0^{(t)} \right\rangle_t$ and $\bar{\theta}_0^{(t)} = \theta'_0{}^{(t)} - \left\langle \theta'_0{}^{(t)} \right\rangle_t + \left\langle \theta_0^{(t)} \right\rangle_t$ (*Figure 4—figure supplement 1*). Finally, we defined the worm's behavior classification as turn when $\bar{\theta}_0^{(t)} < -0.07$, as forward locomotion when $\bar{\theta}_0^{(t)} \geq -0.07$ and $\frac{\mathrm{d}\bar{\xi}_0^{(t)}}{\mathrm{d}t} > 0$, and as backward locomotion when $\bar{\theta}_0^{(t)} \geq -0.07$ and $\frac{\mathrm{d}\bar{\xi}_0^{(t)}}{\mathrm{d}t} \leq 0$ (*Figure 4—figure supplement 1*).

## Acknowledgements

This work is supported by the p-CoE program of DGIST 23-CoE-BT-01. We also acknowledge Prof. Kyuhyung Kim for fruitful discussions on experiments of *C.elegans*.

## Additional information

### Funding

| Funder | Grant reference number | Author |
|---|---|---|
| The p-CoE program of DGIST | 23-CoE-BT-01 | Iksoo Chang |

The funders had no role in study design, data collection and interpretation, or the decision to submit the work for publication.

### Author contributions

Taegon Chung, Conceptualization, Resources, Data curation, Software, Formal analysis, Validation, Investigation, Visualization, Methodology, Writing - original draft, Writing - review and editing; Iksoo Chang, Formal analysis, Supervision, Funding acquisition, Writing - original draft, Project administration, Writing - review and editing; Sangyeol Kim, Conceptualization, Formal analysis, Validation, Investigation, Methodology, Writing - original draft, Writing - review and editing

### Author ORCIDs

Taegon Chung ⦿ http://orcid.org/0009-0004-2335-3500
Sangyeol Kim ⦿ http://orcid.org/0009-0001-3200-9726

Reviewer #1 (Public Review): https://doi.org/10.7554/eLife.92562.3.sa1
Reviewer #2 (Public Review): https://doi.org/10.7554/eLife.92562.3.sa2
Reviewer #3 (Public Review): https://doi.org/10.7554/eLife.92562.3.sa3
Author response https://doi.org/10.7554/eLife.92562.3.sa4

## Additional files

### Supplementary files
• MDAR checklist

### Data availability
All data generated or analysed during this study are included in the manuscript and supporting files. The source code of the program used in this study can be obtained from the open database GitHub (https://github.com/taegonchung/elegansbot, copy archived at *Chung, 2024*) or Python software repository PyPI (https://pypi.org/project/ElegansBot/).

The following previously published dataset was used:

| Author(s) | Year | Dataset title | Dataset URL | Database and Identifier |
|---|---|---|---|---|
| Broekmans OD, Rodgers JB, Ryu WS, Stephens GJ | 2016 | Data from: Resolving coiled shapes reveals new reorientation behaviors in *C. elegans* | http://dx.doi.org/10.5061/dryad.t0m6p | Dryad Digital Repository, 10.5061/dryad.t0m6p |

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

## Appendix 1

### Worm's mass, actuator elasticity coefficient, and damping coefficient

### 1. Mass

The maximum radius of the worm is 40μm (**Boyle et al., 2012**). Assuming that the border surrounding the cross-section parallel to the anterior-posterior axis is a sine function, the average radius is $\gamma = 40 \times 2/\pi \simeq 25\mu m$. Assuming that the worm is a cylindrical body with a bottom surface radius of 25μm, the volume of the worm is $1mm \left(0.025mm\right)^2 \pi \simeq 0.002mm^3$ , and the density of the worm is $\simeq 1000\mu g/mm^3$ (**Reina et al., 2013**), so the weight of the worm is $M = 1000\mu g/mm^3 \times 0.002mm^3 = 2\mu g$.

### 2. Torque elasticity coefficient

In previous research, the muscle elasticity and damping coefficient were designed as functions of the input signal (**Boyle et al., 2012**). The maximum value of this muscle elasticity coefficient is $k_{max} = 2.8 \cdot 10^8 \left[\mu g/sec^2\right]$, and the maximum value of the muscle damping coefficient is $\left(k_{max}/5.6\right) \cdot \left(1sec\right)$ . When $\theta_{i-1} = \theta_i = \theta_{i+1} = 0$ and the length of the moment arm where the muscle exerts force is equal to the average radius $\gamma$ of the worm, the change in the elastic torque due to the change in $\theta_i$ is the torque elasticity coefficient $\kappa$, so $\kappa = \frac{d\tau_{\kappa,i}}{d\theta_i} = \frac{d}{d\theta_i} \left(\gamma \left(k \left(2\gamma\tan\left(\theta_i/2\right)\right)\right)\right) \simeq k\gamma^2 = 1.75 \cdot 10^5 \mu g \cdot mm^2/ \left(sec^2 \cdot rad\right)$. In ElegansBot, it was assumed that this $\kappa$ value is constant regardless of $\theta_{i-1}$ , $\theta_i$ , $\theta_{i+1}$ . In the same way, the torque muscle damping coefficient is $c = \left(1/5.6\right) \cdot 1.75 \cdot 10^5 \mu g \cdot mm^2/ \left(sec \cdot rad\right)$ .

### Minimum information required to describe the motion of each rod

### 1. Information required to describe the movement of the rod

To describe the motion of all rods, it is necessary to know the position ($\mathbf{d}_i$), velocity ($\mathbf{v}_i$), angle ($s_i$) measured counterclockwise from the positive x-axis to the direction of vector $\mathbf{r}_i$ , and angular velocity($\omega_i$) of every i-rod. However, knowing only the position and velocity of the worm ($\mathbf{d}_c$ , $\mathbf{v}_c$) and the angle and angular velocity of every i-rod ($s_i$ , $\omega_i$) is sufficient to describe the motion of all rods.

### 2. Boundary conditions for the position of the rod

The center of mass of the worm is $\mathbf{d}_c = \begin{bmatrix} x_c & y_c \end{bmatrix}^T = \left(\sum_{i=1}^n m\mathbf{d}_i\right)/M = \left(\sum_{i=1}^n \mathbf{d}_i\right)/n$, and the vector parallel to i-rod is $\mathbf{r}_i \equiv r \begin{bmatrix} \cos(s_i) & \sin(s_i) \end{bmatrix}^T = r\widehat{\mathbf{r}}_i$ . If the position vector where i-rod and (i+1)-rod meet is $\mathbf{d}_{tip_i}$ , and the free end of 1-rod and n-rod are $\mathbf{d}_{tip_0}$ and $\mathbf{d}_{tip_n}$ , then $\mathbf{d}_{tip_i} = \mathbf{d}_i + \mathbf{r}_i = \mathbf{d}_{i+1} - \mathbf{r}_{i+1}$ (**Appendix 1—figure 1A**).

### 3. Method of calculating the relative and absolute position of the rod

If the relative coordinate with $\mathbf{d}_{tip_0}$ as the origin is designated as $\mathbf{d}'$, then the following equations satisfy.

$$\mathbf{d}'_{tip_0} = \begin{bmatrix} 0 \\ 0 \end{bmatrix}$$

$$\mathbf{d}'_{tip_i} = \sum_{j=1}^i 2\mathbf{r}_j$$

$$\mathbf{d}'_i = \frac{\mathbf{d}'_{tip_i} + \mathbf{d}'_{tip_{i-1}}}{2}$$

$$\mathbf{d}'_c = \frac{1}{n} \sum_{i=1}^n \mathbf{d}'_i$$

$$\mathbf{d}_i = \mathbf{d}'_i - \mathbf{d}'_c + \mathbf{d}_c$$

Thus, $\mathbf{d}_i$ can be calculated from $\mathbf{d}_c$ and $s_i$ (**Appendix 1—figure 1B**).

### 4. Method of calculating the relative and absolute velocity of the rod

Based on the relationship between the worm's momentum and the rods' momentum, $\mathbf{v}_c = \left(\sum_{i=1}^n m\mathbf{v}_i\right)/M = \left(\sum_{i=1}^n \mathbf{v}_i\right)/n$. In the same way as the location information compression, the relative velocity vector $\mathbf{v}'$ with $\mathbf{v}_{tip_0}$ as the origin has the following relationships ($\widehat{\mathbf{z}} \equiv \widehat{\mathbf{r}}_i \times \widehat{\mathbf{N}}_i$).

$$\mathbf{v}'_{\text{tip}_0} = \begin{bmatrix} 0 \\ 0 \end{bmatrix}$$

$$\mathbf{v}'_{\text{tip}_i} = \sum_{j=1}^{i} \omega_j \hat{\mathbf{z}} \times (2\mathbf{r}_j)$$

$$\mathbf{v}'_i = \frac{\mathbf{v}'_{\text{tip}_i} + \mathbf{v}'_{\text{tip}_{i-1}}}{2}$$

$$\mathbf{v}'_c = \frac{1}{n} \sum_{i=1}^{n} \mathbf{v}'_i$$

$$\mathbf{v}_i = \mathbf{v}'_i - \mathbf{v}'_c + \mathbf{v}_c$$

Thus, $\mathbf{v}_i$ can be calculated from $\mathbf{v}_c$ , $s_i$ , and $\omega_i$ (*Appendix 1—figure 1C*).

## Preservation of linearity in friction

The velocity $\mathbf{v}$ of an arbitrary point particle which is included by i-rod can be decomposed into two velocity components $\mathbf{v} = \mathbf{v}_\alpha + \mathbf{v}_\beta$ . In this case, if the object is subject to an anisotropic Stokes friction, there is linearity between the friction $\mathbf{F}_{b,\alpha}$ , $\mathbf{F}_{b,\beta}$ obtained from each velocity component $\mathbf{v}_\alpha$ , $\mathbf{v}_\beta$ and the friction $\mathbf{F}_b$ obtained from velocity $\mathbf{v}$.

$$\mathbf{F}_{b,\alpha} = -b_\parallel n^{-1}(\mathbf{v}_\alpha \cdot \hat{\mathbf{r}}_i)\hat{\mathbf{r}}_i - b_\perp n^{-1}(\mathbf{v}_\alpha \cdot \hat{\mathbf{N}}_i)\hat{\mathbf{N}}_i$$

$$\mathbf{F}_{b,\beta} = -b_\parallel n^{-1}(\mathbf{v}_\beta \cdot \hat{\mathbf{r}}_i)\hat{\mathbf{r}}_i - b_\perp n^{-1}(\mathbf{v}_\beta \cdot \hat{\mathbf{N}}_i)\hat{\mathbf{N}}_i$$

$$\begin{aligned}
\mathbf{F}_b &= -b_\parallel n^{-1}(\mathbf{v} \cdot \hat{\mathbf{r}}_i)\hat{\mathbf{r}}_i - b_\perp n^{-1}(\mathbf{v} \cdot \hat{\mathbf{N}}_i)\hat{\mathbf{N}}_i \\
&= -b_\parallel n^{-1}\left((\mathbf{v}_\alpha + \mathbf{v}_\beta) \cdot \hat{\mathbf{r}}_i\right)\hat{\mathbf{r}}_i - b_\perp n^{-1}\left((\mathbf{v}_\alpha + \mathbf{v}_\beta) \cdot \hat{\mathbf{N}}_i\right)\hat{\mathbf{N}}_i \\
&= -b_\parallel n^{-1}(\mathbf{v}_\alpha \cdot \hat{\mathbf{r}}_i)\hat{\mathbf{r}}_i - b_\perp n^{-1}(\mathbf{v}_\alpha \cdot \hat{\mathbf{N}}_i)\hat{\mathbf{N}}_i \\
&\quad -b_\parallel n^{-1}(\mathbf{v}_\beta \cdot \hat{\mathbf{r}}_i)\hat{\mathbf{r}}_i - b_\perp n^{-1}(\mathbf{v}_\beta \cdot \hat{\mathbf{N}}_i)\hat{\mathbf{N}}_i \\
&= \mathbf{F}_{b,\alpha} + \mathbf{F}_{b,\beta}
\end{aligned}$$

## Frictional torque by rotational motion

Let us denote variable $\rho$ as the distance from the center of i-rod measured along the direction of vector $\hat{\mathbf{r}}_i$ . It is to be noted that $\rho$ is within the range $[-r, r]$ . For an infinitesimal $\mathrm{d}\rho$ where $0 < \mathrm{d}\rho \ll 1$, the moment arm vector for the infinitesimal interval $[\rho - \mathrm{d}\rho/2, \rho + \mathrm{d}\rho/2]$ (hereafter referred to as the infinitesimal interval $\rho$) from the center of i-rod is $\rho\hat{\mathbf{r}}_i$ . The coefficient of friction for the infinitesimal interval $\rho$ is:

$$\frac{b_\perp}{n} \frac{\mathrm{d}\rho}{2r}$$

The velocity component due to the rotational motion of the infinitesimal interval $\rho$ is $\rho\omega_i\hat{\mathbf{N}}_i$ . Therefore, the frictional force received by the infinitesimal interval $\rho$ due to rotational motion is:

$$-\frac{b_\perp}{n} \frac{\mathrm{d}\rho}{2r} \rho\omega_i\hat{\mathbf{N}}_i = -\frac{1}{2}\frac{b_\perp}{nr}\omega_i\rho\mathrm{d}\rho\hat{\mathbf{N}}_i$$

The torque received by the infinitesimal interval $\rho$ due to rotational motion is:

$$\begin{aligned}
\mathrm{d}\boldsymbol{\tau} &= (\rho\hat{\mathbf{r}}_i) \times (-\frac{1}{2}\frac{b_\perp}{nr}\omega_i\rho\mathrm{d}\rho\hat{\mathbf{N}}_i) \\
&= -\frac{1}{2}\frac{b_\perp}{nr}\omega_i\rho^2\mathrm{d}\rho\hat{\mathbf{r}}_i \times \hat{\mathbf{N}}_i \\
&= -\frac{1}{2}\frac{b_\perp}{nr}\omega_i\rho^2\mathrm{d}\rho\hat{\mathbf{z}}
\end{aligned}$$

The total frictional torque received by i-rod due to rotational motion is

$$
\begin{aligned}
\tau_{b,i} &= \int_{-r}^{r} d\tau \\
&= \int_{-r}^{r} -\frac{1}{2}\frac{b_\perp}{nr}\omega_i\rho^2 d\rho\hat{\mathbf{z}} \\
&= -\frac{1}{2}\frac{b_\perp}{nr}\omega_i\left[\frac{1}{3}\rho^3\right]_{-r}^{r}\hat{\mathbf{z}} \\
&= -\frac{1}{2}\frac{b_\perp}{nr}\omega_i\frac{2}{3}r^3\hat{\mathbf{z}} \\
&= -\frac{1}{3}\frac{b_\perp}{n}r^2\omega_i\hat{\mathbf{z}}
\end{aligned}
$$

## Proof of muscle force

The muscle force, which makes the torque $\tau_i$ received from i-actuator to i-rod, was designed as follows. The damped torsion spring is connected at the center of each rod and gives a force in a direction perpendicular to the rod. The support is connected to i-actuator's midpoint and i-joint (*Appendix 1—figure 2A*). Let us say that the mass of the support and i-actuator are both 0. The support gives forces ($\mathbf{F}_{\mathrm{sup}_1,i}$, $\mathbf{F}_{\mathrm{sup}_2,i}$) of the same size to i-rod and (i+1)-rod in a direction parallel to the support (*Appendix 1—figure 2B*). Let us assume that the resultant force received by i-actuator is **0** (*Appendix 1—figure 2C*). Thus, the forces from the actuator generate no torque at the connection point in the middle of the rod but at the connection point at the end (*Appendix 1—figure 2D*).

$$
\begin{aligned}
F_{\mathrm{sup}_1,i} &= F_{\mathrm{p}_1,i}\cos\frac{\theta_i}{2} \\
\tau_i &= \left(\mathbf{r}_i \times \mathbf{F}_{\mathrm{sup}_1,i}\right)\cdot\hat{\mathbf{z}} \\
&= rF_{\mathrm{sup}_1,i}\cos\frac{\theta_i}{2} \\
F_{\mathrm{sup}_1,i} &= \frac{\tau_i}{r\cos\dfrac{\theta_i}{2}} \\
F_{\mathrm{p}_1,i} &= \frac{F_{\mathrm{sup}_1,i}}{\cos\dfrac{\theta_i}{2}} \\
&= \frac{\tau_i}{r\cos^2\dfrac{\theta_i}{2}}
\end{aligned}
$$

The force that i-rod receives from i-actuator is $\mathbf{F}_{\mathrm{p}_1,i} + \mathbf{F}_{\mathrm{sup}_1,i}$ and the magnitude of the force is $\left|\mathbf{F}_{\mathrm{p}_1,i} + \mathbf{F}_{\mathrm{sup}_1,i}\right| = F_{\mathrm{p}_1,i}\sin\left(\theta_i/2\right) = \tau_i\sin\left(\theta_i/2\right)/\left(r\cos^2\left(\theta_i/2\right)\right)$ and the direction of the force is $-\left[\cos\left((s_i+s_{i+1})/2\right) \quad \sin\left((s_i+s_{i+1})/2\right)\right]^{\mathsf{T}}$. These equations come down to the following equations.

$$
\mathbf{F}_{\mathrm{p}_1,i} + \mathbf{F}_{\mathrm{sup}_1,i} = -\frac{\tau_i\sin\dfrac{\theta_i}{2}}{r\cos^2\dfrac{\theta_i}{2}}\begin{bmatrix}\cos\left(\dfrac{s_i+s_{i+1}}{2}\right) \\ \sin\left(\dfrac{s_i+s_{i+1}}{2}\right)\end{bmatrix}
$$

$$
\mathbf{F}_{\mathrm{p}_1,i} + \mathbf{F}_{\mathrm{sup}_1,i} + \mathbf{F}_{\mathrm{p}_2,i} + \mathbf{F}_{\mathrm{sup}_2,i} = \mathbf{0}
$$

$$
\mathbf{F}_{\mathrm{p}_2,i} + \mathbf{F}_{\mathrm{sup}_2,i} = \frac{\tau_i\sin\dfrac{\theta_i}{2}}{r\cos^2\dfrac{\theta_i}{2}}\begin{bmatrix}\cos\left(\dfrac{s_i+s_{i+1}}{2}\right) \\ \sin\left(\dfrac{s_i+s_{i+1}}{2}\right)\end{bmatrix}
$$

Therefore, the total force that i-rod receives from i-actuator and (i-1)-actuator is

$$
\mathbf{F}_{c\kappa,i} = \sum_{j=-1}^{0}\frac{(-1)^{j-1}\tau_{i+j}\sin\dfrac{\theta_{i+j}}{2}}{r\cos^2\dfrac{\theta_{i+j}}{2}}\begin{bmatrix}\cos\left(\dfrac{s_{i+j}+s_{i+j+1}}{2}\right) \\ \sin\left(\dfrac{s_{i+j}+s_{i+j+1}}{2}\right)\end{bmatrix}
$$

## Joint force calculation method

Let us calculate the joint force $\mathbf{F}_i$ from the given values ($s_i$, $\mathbf{F}_{c\kappa,i}$, $\mathbf{F}_{b,i}$, $\tau_{c\kappa,i}$, $\tau_{b,i}$). In this part, the superscript $*^T$ means the transpose of a vector or a matrix. i-rod and (i+1)-rod always meet at i-joint can be expressed as a following vector equation.

$$\mathbf{d}_i + \mathbf{r}_i = \mathbf{d}_{i+1} - \mathbf{r}_{i+1}$$

Differentiating this equation twice for time, we can see that the accelerations of the ends of i-rod and (i+1)-rod at i-joint are the same.

$$
\begin{aligned}
\frac{d^2}{dt^2}(\mathbf{d}_i + \mathbf{r}_i) &= \frac{d^2}{dt^2}(\mathbf{d}_i + r\widehat{\mathbf{r}}_i) \\
&= \frac{d}{dt}(\mathbf{v}_i + r\omega_i\widehat{\mathbf{N}}_i) \\
&= \mathbf{a}_i + r\frac{d\omega_i}{dt}\widehat{\mathbf{N}}_i + r\omega_i\frac{d\widehat{\mathbf{N}}_i}{dt} \\
&= \mathbf{a}_i + r\alpha_i\widehat{\mathbf{N}}_i - r\omega_i^2\widehat{\mathbf{r}}_i \\
&= \mathbf{a}_i + \boldsymbol{\alpha}_i \times \mathbf{r}_i - \omega_i^2\mathbf{r}_i
\end{aligned}
$$

$$
\begin{aligned}
\frac{d^2}{dt^2}(\mathbf{d}_{i+1} - \mathbf{r}_{i+1}) &= \frac{d^2}{dt^2}(\mathbf{d}_{i+1} - r\widehat{\mathbf{r}}_{i+1}) \\
&= \frac{d}{dt}(\mathbf{v}_{i+1} - r\omega_{i+1}\widehat{\mathbf{N}}_{i+1}) \\
&= \mathbf{a}_{i+1} - r\frac{d\omega_{i+1}}{dt}\widehat{\mathbf{N}}_{i+1} - r\omega_{i+1}\frac{d\widehat{\mathbf{N}}_{i+1}}{dt} \\
&= \mathbf{a}_{i+1} - r\alpha_{i+1}\widehat{\mathbf{N}}_{i+1} + r\omega_{i+1}^2\widehat{\mathbf{r}}_{i+1} \\
&= \mathbf{a}_{i+1} - \boldsymbol{\alpha}_{i+1} \times \mathbf{r}_{i+1} + \omega_{i+1}^2\mathbf{r}_{i+1}
\end{aligned}
$$

$$
\begin{aligned}
\frac{d^2}{dt^2}(\mathbf{d}_i + \mathbf{r}_i) &= \frac{d^2}{dt^2}(\mathbf{d}_{i+1} - \mathbf{r}_{i+1}) \\
&= \mathbf{a}_i + \boldsymbol{\alpha}_i \times \mathbf{r}_i - \omega_i^2\mathbf{r}_i \\
&= \mathbf{a}_{i+1} - \boldsymbol{\alpha}_{i+1} \times \mathbf{r}_{i+1} + \omega_{i+1}^2\mathbf{r}_{i+1}
\end{aligned}
$$

Multiplying both sides by $m$ gives:

$$m\left(\mathbf{a}_i + \boldsymbol{\alpha}_i \times \mathbf{r}_i\right) - m\omega_i^2\mathbf{r}_i = m\left(\mathbf{a}_{i+1} - \boldsymbol{\alpha}_{i+1} \times \mathbf{r}_{i+1}\right) + m\omega_{i+1}^2\mathbf{r}_{i+1}$$

If $\mathbf{F}_{\mathrm{res},i}$ is the total force applied to i-rod other than $\mathbf{F}_i$ and $-\mathbf{F}_{i-1}$, which is $\mathbf{F}_{\mathrm{res},i} = \mathbf{F}_{c\kappa,i} + \mathbf{F}_{b,i}$, then:

$$m\mathbf{a}_i = \mathbf{F}_i - \mathbf{F}_{i-1} + \mathbf{F}_{\mathrm{res},i}$$

If $\tau_{\mathrm{res},i}$ is the total torque applied to i-rod excluding $\mathbf{r}_i \times \left(\mathbf{F}_{i-1} + \mathbf{F}_i\right)$, which is $\tau_{\mathrm{res},i} = \tau_{c\kappa,i} + \tau_{b,i}$, then:

$$I\alpha_i = \frac{1}{3}mr^2\alpha_i = \mathbf{r}_i \times \left(\mathbf{F}_{i-1} + \mathbf{F}_i\right) + \tau_{\mathrm{res},i}$$

Dividing both sides by $\frac{1}{3}r^2$ gives:

$$m\alpha_i = 3r^{-2}\mathbf{r}_i \times \left(\mathbf{F}_{i-1} + \mathbf{F}_i\right) + 3r^{-2}\tau_{\mathrm{res},i}$$

If $\mathbf{h}_{\mathrm{res},i} \equiv 3r^{-2}\tau_{\mathrm{res},i} \times \mathbf{r}_i$, then:

$$
\begin{aligned}
m\left(\mathbf{a}_i + \boldsymbol{\alpha}_i \times \mathbf{r}_i\right) &= \left(\mathbf{F}_i - \mathbf{F}_{i-1} + \mathbf{F}_{\mathrm{res},i}\right) + \left[3r^{-2}\mathbf{r}_i \times \left(\mathbf{F}_{i-1} + \mathbf{F}_i\right) + 3r^{-2}\tau_{\mathrm{res},i}\right] \times \mathbf{r}_i \\
&= \left(\mathbf{F}_i - \mathbf{F}_{i-1}\right) + 3\left[\widehat{\mathbf{r}}_i \times \left(\mathbf{F}_{i-1} + \mathbf{F}_i\right)\right] \times \widehat{\mathbf{r}}_i + \mathbf{F}_{\mathrm{res},i} + 3r^{-2}\tau_{\mathrm{res},i} \times \mathbf{r}_i \\
&= \left(\mathbf{F}_i - \mathbf{F}_{i-1}\right) + 3\left[\left(\mathbf{F}_{i-1} + \mathbf{F}_i\right) \cdot \widehat{\mathbf{N}}_i\right]\widehat{\mathbf{N}}_i + \mathbf{F}_{\mathrm{res},i} + \mathbf{h}_{\mathrm{res},i}
\end{aligned}
$$

Following the same method,

$$m\left(\mathbf{a}_{i+1} - \boldsymbol{\alpha}_{i+1} \times \mathbf{r}_{i+1}\right) = \left(\mathbf{F}_{i+1} - \mathbf{F}_i\right) - 3\left[\left(\mathbf{F}_i + \mathbf{F}_{i+1}\right) \cdot \widehat{\mathbf{N}}_{i+1}\right]\widehat{\mathbf{N}}_{i+1} + \mathbf{F}_{\mathrm{res},i+1} - \mathbf{h}_{\mathrm{res},i+1}$$

To organize the terms of the equations into known values and unknown values,

$$
\begin{aligned}
\mathbf{k}_i &\equiv \mathbf{F}_i - \mathbf{F}_{i-1} \\
\mathbf{P}_i &\equiv 3\widehat{\mathbf{N}}_i\widehat{\mathbf{N}}_i^{\mathsf{T}} \\
\mathbf{h}_i &\equiv 3\left[\left(\mathbf{F}_{i-1} + \mathbf{F}_i\right) \cdot \widehat{\mathbf{N}}_i\right]\widehat{\mathbf{N}}_i \\
&= 3\widehat{\mathbf{N}}_i\widehat{\mathbf{N}}_i^{\mathsf{T}}\left(\mathbf{F}_{i-1} + \mathbf{F}_i\right) \\
&= \mathbf{P}_i\left(\mathbf{F}_{i-1} + \mathbf{F}_i\right) \\
m\left(\mathbf{a}_i + \boldsymbol{\alpha}_i \times \mathbf{r}_i\right) &= \mathbf{k}_i + \mathbf{h}_i + \mathbf{F}_{\mathrm{res},i} + \mathbf{h}_{\mathrm{res},i} \\
m\left(\mathbf{a}_{i+1} - \boldsymbol{\alpha}_{i+1} \times \mathbf{r}_{i+1}\right) &= \mathbf{k}_{i+1} - \mathbf{h}_{i+1} + \mathbf{F}_{\mathrm{res},i+1} - \mathbf{h}_{\mathrm{res},i+1} \\
\mathbf{k}_i + \mathbf{h}_i + \mathbf{F}_{\mathrm{res},i} + \mathbf{h}_{\mathrm{res},i} - m\omega_i^2\mathbf{r}_i &= \mathbf{k}_{i+1} - \mathbf{h}_{i+1} + \mathbf{F}_{\mathrm{res},i+1} - \mathbf{h}_{\mathrm{res},i+1} + m\omega_{i+1}^2\mathbf{r}_{i+1} \\
\mathbf{q}_i &\equiv \mathbf{k}_i + \mathbf{h}_i - \mathbf{k}_{i+1} + \mathbf{h}_{i+1} \\
&= -\mathbf{F}_{\mathrm{res},i} - \mathbf{h}_{\mathrm{res},i} + \mathbf{F}_{\mathrm{res},i+1} - \mathbf{h}_{\mathrm{res},i+1} + m\omega_i^2\mathbf{r}_i + m\omega_{i+1}^2\mathbf{r}_{i+1}
\end{aligned}
$$

Therefore, if we know all $\mathbf{F}_{\mathrm{res},i}$ and $\tau_{\mathrm{res},i}$ for each i, we can find $\mathbf{q}_i$. If $\mathbf{I} \equiv \begin{bmatrix} 1 & 0 \\ 0 & 1 \end{bmatrix}$, and if we expand

$\mathbf{q}_i$,

$$
\begin{aligned}
\mathbf{q}_i &\equiv \mathbf{k}_i + \mathbf{h}_i - \mathbf{k}_{i+1} + \mathbf{h}_{i+1} \\
&= -\mathbf{k}_{i+1} + \mathbf{k}_i + \mathbf{h}_i + \mathbf{h}_{i+1} \\
&= -\mathbf{F}_{i+1} + \mathbf{F}_i + \mathbf{F}_i - \mathbf{F}_{i-1} + \mathbf{P}_i\left(\mathbf{F}_{i-1} + \mathbf{F}_i\right) + \mathbf{P}_{i+1}\left(\mathbf{F}_i + \mathbf{F}_{i+1}\right) \\
&= \mathbf{F}_{i-1}\left(\mathbf{P}_i - \mathbf{I}\right) + \mathbf{F}_i\left(\mathbf{P}_i + \mathbf{P}_{i+1} + 2\mathbf{I}\right) + \mathbf{F}_{i+1}\left(\mathbf{P}_{i+1} - \mathbf{I}\right)
\end{aligned}
$$

If we set $\mathbf{A}_i \equiv \mathbf{P}_i - \mathbf{I}$ and $\mathbf{B}_i \equiv \mathbf{P}_i + \mathbf{P}_{i+1} + 2\mathbf{I}$, then:

$$\mathbf{q}_i = \mathbf{A}_i\mathbf{F}_{i-1} + \mathbf{B}_i\mathbf{F}_i + \mathbf{A}_{i+1}\mathbf{F}_{i+1}$$

If we set $\overline{\mathbf{0}} = \begin{bmatrix} 0 & 0 \\ 0 & 0 \end{bmatrix}$ and express the above equation in a multidimensional tensor form,

$$
\begin{bmatrix}
\mathbf{B}_1 & \mathbf{A}_2 & \overline{\mathbf{0}} & \overline{\mathbf{0}} & \overline{\mathbf{0}} & \cdots & \overline{\mathbf{0}} \\
\mathbf{A}_2 & \mathbf{B}_2 & \mathbf{A}_3 & \overline{\mathbf{0}} & \overline{\mathbf{0}} & \cdots & \overline{\mathbf{0}} \\
\overline{\mathbf{0}} & \mathbf{A}_3 & \mathbf{B}_3 & \mathbf{A}_4 & \overline{\mathbf{0}} & \cdots & \overline{\mathbf{0}} \\
\overline{\mathbf{0}} & \overline{\mathbf{0}} & \mathbf{A}_4 & \mathbf{B}_4 & \mathbf{A}_5 & \cdots & \overline{\mathbf{0}} \\
\overline{\mathbf{0}} & \overline{\mathbf{0}} & \overline{\mathbf{0}} & \mathbf{A}_5 & \mathbf{B}_5 & \ddots & \vdots \\
\vdots & \vdots & \vdots & \vdots & \ddots & \ddots & \mathbf{A}_{n-1} \\
\overline{\mathbf{0}} & \overline{\mathbf{0}} & \overline{\mathbf{0}} & \overline{\mathbf{0}} & \cdots & \mathbf{A}_{n-1} & \mathbf{B}_{n-1}
\end{bmatrix}
\begin{bmatrix}
\mathbf{F}_1 \\ \mathbf{F}_2 \\ \mathbf{F}_3 \\ \mathbf{F}_4 \\ \vdots \\ \mathbf{F}_{n-2} \\ \mathbf{F}_{n-1}
\end{bmatrix}
=
\begin{bmatrix}
\mathbf{q}_1 \\ \mathbf{q}_2 \\ \mathbf{q}_3 \\ \mathbf{q}_4 \\ \vdots \\ \mathbf{q}_{n-2} \\ \mathbf{q}_{n-1}
\end{bmatrix}
$$

Let us set:

$$
\mathbf{D} \equiv
\begin{bmatrix}
\mathbf{B}_1 & \mathbf{A}_2 & \overline{\mathbf{0}} & \overline{\mathbf{0}} & \overline{\mathbf{0}} & \cdots & \overline{\mathbf{0}} \\
\mathbf{A}_2 & \mathbf{B}_2 & \mathbf{A}_3 & \overline{\mathbf{0}} & \overline{\mathbf{0}} & \cdots & \overline{\mathbf{0}} \\
\overline{\mathbf{0}} & \mathbf{A}_3 & \mathbf{B}_3 & \mathbf{A}_4 & \overline{\mathbf{0}} & \cdots & \overline{\mathbf{0}} \\
\overline{\mathbf{0}} & \overline{\mathbf{0}} & \mathbf{A}_4 & \mathbf{B}_4 & \mathbf{A}_5 & \cdots & \overline{\mathbf{0}} \\
\overline{\mathbf{0}} & \overline{\mathbf{0}} & \overline{\mathbf{0}} & \mathbf{A}_5 & \mathbf{B}_5 & \ddots & \vdots \\
\vdots & \vdots & \vdots & \vdots & \ddots & \ddots & \mathbf{A}_{n-1} \\
\overline{\mathbf{0}} & \overline{\mathbf{0}} & \overline{\mathbf{0}} & \overline{\mathbf{0}} & \cdots & \mathbf{A}_{n-1} & \mathbf{B}_{n-1}
\end{bmatrix}
$$

$$
F \equiv \begin{bmatrix} \mathbf{F}_1 \\ \mathbf{F}_2 \\ \mathbf{F}_3 \\ \mathbf{F}_4 \\ \vdots \\ \mathbf{F}_{n-2} \\ \mathbf{F}_{n-1} \end{bmatrix}
$$

$$
Q \equiv \begin{bmatrix} \mathbf{q}_1 \\ \mathbf{q}_2 \\ \mathbf{q}_3 \\ \mathbf{q}_4 \\ \vdots \\ \mathbf{q}_{n-2} \\ \mathbf{q}_{n-1} \end{bmatrix}
$$

As $\mathbf{A}_i$ and $\mathbf{B}_i$ can be calculated from $\mathbf{P}_i$, $\mathbf{P}_i$ from $\widehat{\mathbf{N}}_i$, and $\widehat{\mathbf{N}}_i$ from $s_i$, we can find $\mathcal{D}$ from $s_i$.

$$
\begin{aligned}
\mathbf{q}_i &= -\mathbf{F}_{\mathrm{res},i} + \mathbf{F}_{\mathrm{res},i+1} - \mathbf{h}_{\mathrm{res},i} - \mathbf{h}_{\mathrm{res},i+1} + m\omega_i^2 \mathbf{r}_i + m\omega_{i+1}^2 \mathbf{r}_{i+1} \\
&= -\mathbf{F}_{\mathrm{res},i} + \mathbf{F}_{\mathrm{res},i+1} - 3r^{-2}\left(\boldsymbol{\tau}_{\mathrm{res},i} \times \mathbf{r}_i + \boldsymbol{\tau}_{\mathrm{res},i+1} \times \mathbf{r}_{i+1}\right) + m\omega_i^2 \mathbf{r}_i + m\omega_{i+1}^2 \mathbf{r}_{i+1}
\end{aligned}
$$

Therefore, if we know $\mathbf{F}_{c\kappa,i}$, $\mathbf{F}_{b,i}$, $\tau_{c\kappa,i}$, $\tau_{b,i}$, we can find $\mathcal{Q}$.

$$
\therefore \mathcal{D}\mathcal{F} = \mathcal{Q}
$$

We can find $\mathcal{F}$, thus $\mathbf{F}_i$ ($i \in \{1, \cdots, n-1\}$), by solving this tensor equation.

If we denote each component of the tensor, which are each matrix and the p-th row q-th column component of the vector, as $(*)_{p,q}$, then:

$$
\begin{bmatrix}
\begin{bmatrix} (\mathbf{B}_1)_{1,1} & (\mathbf{B}_1)_{1,2} \\ (\mathbf{B}_1)_{2,1} & (\mathbf{B}_1)_{2,2} \end{bmatrix} & \begin{bmatrix} (\mathbf{A}_2)_{1,1} & (\mathbf{A}_2)_{1,2} \\ (\mathbf{A}_2)_{2,1} & (\mathbf{A}_2)_{2,2} \end{bmatrix} & \begin{bmatrix} 0 & 0 \\ 0 & 0 \end{bmatrix} & \cdots \\[2em]
\begin{bmatrix} (\mathbf{A}_2)_{1,1} & (\mathbf{A}_2)_{1,2} \\ (\mathbf{A}_2)_{2,1} & (\mathbf{A}_2)_{2,2} \end{bmatrix} & \begin{bmatrix} (\mathbf{B}_2)_{1,1} & (\mathbf{B}_2)_{1,2} \\ (\mathbf{B}_2)_{2,1} & (\mathbf{B}_2)_{2,2} \end{bmatrix} & \begin{bmatrix} (\mathbf{A}_3)_{1,1} & (\mathbf{A}_3)_{1,2} \\ (\mathbf{A}_3)_{2,1} & (\mathbf{A}_3)_{2,2} \end{bmatrix} & \cdots \\[2em]
\begin{bmatrix} 0 & 0 \\ 0 & 0 \end{bmatrix} & \begin{bmatrix} (\mathbf{A}_3)_{1,1} & (\mathbf{A}_3)_{1,2} \\ (\mathbf{A}_3)_{2,1} & (\mathbf{A}_3)_{2,2} \end{bmatrix} & \begin{bmatrix} (\mathbf{B}_3)_{1,1} & (\mathbf{B}_3)_{1,2} \\ (\mathbf{B}_3)_{2,1} & (\mathbf{B}_3)_{2,2} \end{bmatrix} & \cdots \\[2em]
\vdots & \vdots & \vdots & \ddots
\end{bmatrix}
\begin{bmatrix}
\begin{bmatrix} (\mathbf{F}_1)_{1,1} \\ (\mathbf{F}_1)_{2,1} \end{bmatrix} \\[1.5em]
\begin{bmatrix} (\mathbf{F}_2)_{1,1} \\ (\mathbf{F}_2)_{2,1} \end{bmatrix} \\[1.5em]
\begin{bmatrix} (\mathbf{F}_3)_{1,1} \\ (\mathbf{F}_3)_{2,1} \end{bmatrix} \\[1.5em]
\vdots
\end{bmatrix}
=
\begin{bmatrix}
\begin{bmatrix} (\mathbf{q}_1)_{1,1} \\ (\mathbf{q}_1)_{2,1} \end{bmatrix} \\[1.5em]
\begin{bmatrix} (\mathbf{q}_2)_{1,1} \\ (\mathbf{q}_2)_{2,1} \end{bmatrix} \\[1.5em]
\begin{bmatrix} (\mathbf{q}_3)_{1,1} \\ (\mathbf{q}_3)_{2,1} \end{bmatrix} \\[1.5em]
\vdots
\end{bmatrix}
$$

And the matrix equation equivalent to this tensor equation is:

$$
\begin{bmatrix}
(\mathbf{B}_1)_{1,1} & (\mathbf{B}_1)_{1,2} & (\mathbf{A}_2)_{1,1} & (\mathbf{A}_2)_{1,2} & 0 & 0 & \cdots \\
(\mathbf{B}_1)_{2,1} & (\mathbf{B}_1)_{2,2} & (\mathbf{A}_2)_{2,1} & (\mathbf{A}_2)_{2,2} & 0 & 0 & \cdots \\
(\mathbf{A}_2)_{1,1} & (\mathbf{A}_2)_{1,2} & (\mathbf{B}_2)_{1,1} & (\mathbf{B}_2)_{1,2} & (\mathbf{A}_3)_{1,1} & (\mathbf{A}_3)_{1,2} & \cdots \\
(\mathbf{A}_2)_{2,1} & (\mathbf{A}_2)_{2,2} & (\mathbf{B}_2)_{2,1} & (\mathbf{B}_2)_{2,2} & (\mathbf{A}_3)_{2,1} & (\mathbf{A}_3)_{2,2} & \cdots \\
0 & 0 & (\mathbf{A}_3)_{1,1} & (\mathbf{A}_3)_{1,2} & (\mathbf{B}_3)_{1,1} & (\mathbf{B}_3)_{1,2} & \cdots \\
0 & 0 & (\mathbf{A}_3)_{2,1} & (\mathbf{A}_3)_{2,2} & (\mathbf{B}_3)_{2,1} & (\mathbf{B}_3)_{2,2} & \cdots \\
\vdots & \vdots & \vdots & \vdots & \vdots & \vdots & \ddots
\end{bmatrix}
\begin{bmatrix}
(\mathbf{F}_1)_{1,1} \\
(\mathbf{F}_1)_{2,1} \\
(\mathbf{F}_2)_{1,1} \\
(\mathbf{F}_2)_{2,1} \\
(\mathbf{F}_3)_{1,1} \\
(\mathbf{F}_3)_{2,1} \\
\vdots
\end{bmatrix}
=
\begin{bmatrix}
(\mathbf{q}_1)_{1,1} \\
(\mathbf{q}_1)_{2,1} \\
(\mathbf{q}_2)_{1,1} \\
(\mathbf{q}_2)_{2,1} \\
(\mathbf{q}_3)_{1,1} \\
(\mathbf{q}_3)_{2,1} \\
\vdots
\end{bmatrix}
$$

Let us set:

$$
\mathbf{D} \equiv
\begin{bmatrix}
(\mathbf{B}_1)_{1,1} & (\mathbf{B}_1)_{1,2} & (\mathbf{A}_2)_{1,1} & (\mathbf{A}_2)_{1,2} & 0 & 0 & \cdots \\
(\mathbf{B}_1)_{2,1} & (\mathbf{B}_1)_{2,2} & (\mathbf{A}_2)_{2,1} & (\mathbf{A}_2)_{2,2} & 0 & 0 & \cdots \\
(\mathbf{A}_2)_{1,1} & (\mathbf{A}_2)_{1,2} & (\mathbf{B}_2)_{1,1} & (\mathbf{B}_2)_{1,2} & (\mathbf{A}_3)_{1,1} & (\mathbf{A}_3)_{1,2} & \cdots \\
(\mathbf{A}_2)_{2,1} & (\mathbf{A}_2)_{2,2} & (\mathbf{B}_2)_{2,1} & (\mathbf{B}_2)_{2,2} & (\mathbf{A}_3)_{2,1} & (\mathbf{A}_3)_{2,2} & \cdots \\
0 & 0 & (\mathbf{A}_3)_{1,1} & (\mathbf{A}_3)_{1,2} & (\mathbf{B}_3)_{1,1} & (\mathbf{B}_3)_{1,2} & \cdots \\
0 & 0 & (\mathbf{A}_3)_{2,1} & (\mathbf{A}_3)_{2,2} & (\mathbf{B}_3)_{2,1} & (\mathbf{B}_3)_{2,2} & \cdots \\
\vdots & \vdots & \vdots & \vdots & \vdots & \vdots & \ddots
\end{bmatrix}
$$

Since $\mathbf{A}_i$, $\mathbf{B}_i$ are symmetric matrices, $\mathbf{D}$ is a heptadiagonal symmetric matrix. Since $\mathbf{D}$ is a symmetric matrix, the solution to the matrix equation can be found with the Cholesky decomposition. Therefore, we can find the x-axis and y-axis components of $\mathbf{F}_i$.

As a result, we can find the joint force $\mathbf{F}_i$ from the known values ($s_i$, $\mathbf{F}_{c\kappa,i}$, $\mathbf{F}_{b,i}$, $\tau_{c\kappa,i}$, $\tau_{b,i}$).

## Proof of numerical integration for the translational motion of a worm using semi-implicit Euler method

When the friction coefficients $b_\perp$, $b_\parallel$ are sufficiently large compared to $M/\Delta t$, numerical integration via the explicit Euler method ($\mathbf{v}_c^{(t+\Delta t)} = \mathbf{v}_c^{(t)} + \mathbf{a}_c^{(t)}\Delta t = \mathbf{v}_c^{(t)} + \frac{\sum_i \mathbf{F}_{b,i}^{(t)}}{M}\Delta t$) becomes unstable (*Butcher, 2003*). Therefore, for all friction coefficients greater than or equal to 0, the semi-implicit Euler method ($\mathbf{v}_c^{(t+\Delta t)} = \mathbf{v}_c^{(t)} + \mathbf{a}_c^{(t+\Delta t)}\Delta t \simeq \mathbf{v}_c^{(t)} + \frac{1}{1+\frac{b_\perp \Delta t}{M}} \frac{\sum_i \mathbf{F}_{b,i}^{(t)}}{M}\Delta t$) was used to ensure numerical integration remains stable, and its proof is as follows.

Newton's equation for the translational motion of each i-rod is as follows.

$$
\begin{aligned}
\mathbf{F}_i &= m\mathbf{a}_i \\
&= \mathbf{F}_{b,i} + \mathbf{F}_{c\kappa,i} + \mathbf{F}_{\text{joint},i} \\
&= -\frac{b_\perp}{n}\mathbf{v}_{\perp,i} - \frac{b_\parallel}{n}\mathbf{v}_{\parallel,i} + \mathbf{F}_{c\kappa,i} + \mathbf{F}_{\text{joint},i}
\end{aligned}
$$

The integration formula for $\mathbf{a}_i$ using the implicit Euler method is as follows. (where $b_\perp \geq b_\parallel$)

$$
\begin{aligned}
\mathbf{v}_i^{(t+\Delta t)} &= \mathbf{v}_i^{(t)} + \mathbf{a}_i^{(t+\Delta t)}\Delta t \\
&= \mathbf{v}_i^{(t)} - \frac{\Delta t}{m}\left(\frac{b_\perp}{n}\mathbf{v}_{\perp,i}^{(t+\Delta t)} + \frac{b_\parallel}{n}\mathbf{v}_{\parallel,i}^{(t+\Delta t)}\right) + \frac{\Delta t}{m}\left(\mathbf{F}_{c\kappa,i}^{(t+\Delta t)} + \mathbf{F}_{\text{joint},i}^{(t+\Delta t)}\right) \\
&= \mathbf{v}_i^{(t)} - \frac{\Delta t}{M}\left(b_\perp \mathbf{v}_{\perp,i}^{(t+\Delta t)} + b_\parallel \mathbf{v}_{\parallel,i}^{(t+\Delta t)}\right) + \frac{\Delta t}{m}\left(\mathbf{F}_{c\kappa,i}^{(t+\Delta t)} + \mathbf{F}_{\text{joint},i}^{(t+\Delta t)}\right) \\
&= \mathbf{v}_i^{(t)} - \frac{\Delta t}{M}\left(b_\perp \left(\mathbf{v}_{\perp,i}^{(t+\Delta t)} + \mathbf{v}_{\parallel,i}^{(t+\Delta t)}\right) + (b_\parallel - b_\perp)\mathbf{v}_{\parallel,i}^{(t+\Delta t)}\right) + \frac{\Delta t}{m}\left(\mathbf{F}_{c\kappa,i}^{(t+\Delta t)} + \mathbf{F}_{\text{joint},i}^{(t+\Delta t}\right) \\
&= \mathbf{v}_i^{(t)} - \frac{\Delta t}{M}\left(b_\perp \mathbf{v}_i^{(t+\Delta t)} + (b_\parallel - b_\perp)\mathbf{v}_{\parallel,i}^{(t+\Delta t)}\right) + \frac{\Delta t}{m}\left(\mathbf{F}_{c\kappa,i}^{(t+\Delta t)} + \mathbf{F}_{\text{joint},i}^{(t+\Delta t)}\right) \\
\left(1 + \frac{b_\perp \Delta t}{M}\right)\mathbf{v}_i^{(t+\Delta t)} &= \mathbf{v}_i^{(t)} - \frac{\Delta t}{M}\left((b_\parallel - b_\perp)\mathbf{v}_{\parallel,i}^{(t+\Delta t)}\right) + \frac{\Delta t}{m}\left(\mathbf{F}_{c\kappa,i}^{(t+\Delta t)} + \mathbf{F}_{\text{joint},i}^{(t+\Delta t)}\right) \\
&= \mathbf{v}_i^{(t)} + \frac{(b_\perp - b_\parallel)\Delta t}{M}\mathbf{v}_{\parallel,i}^{(t+\Delta t)} + \frac{\Delta t}{m}\left(\mathbf{F}_{c\kappa,i}^{(t+\Delta t)} + \mathbf{F}_{\text{joint},i}^{(t+\Delta t)}\right)
\end{aligned}
$$

Because $\mathbf{v}_{\perp,i}^{(t+\Delta t)}$ and $\mathbf{v}_{\parallel,i}^{(t+\Delta t)}$ are unknown at time $t$, the numerical calculation of the above formula is impossible.

$$\frac{\left| \frac{(b_\perp - b_\parallel)\Delta t}{M} \left( \mathbf{v}_{\parallel,i}^{(t+\Delta t)} - \mathbf{v}_{\parallel,i}^{(t)} \right) \right|}{\left| \mathbf{v}_i^{(t)} + \frac{(b_\perp - b_\parallel)\Delta t}{M} \mathbf{v}_{\parallel,i}^{(t+\Delta t)} + \frac{\Delta t}{m} \left( \mathbf{F}_{c\kappa,i}^{(t+\Delta t)} + \mathbf{F}_{\text{joint},i}^{(t+\Delta t)} \right) \right|} \simeq 0$$

If the above formula is assumed to be true, the following approximation can be used.

$$\left( 1 + \frac{b_\perp \Delta t}{M} \right) \mathbf{v}_i^{(t+\Delta t)} = \mathbf{v}_i^{(t)} + \frac{(b_\perp - b_\parallel)\Delta t}{M} \mathbf{v}_{\parallel,i}^{(t+\Delta t)} + \frac{\Delta t}{m} \left( \mathbf{F}_{c\kappa,i}^{(t+\Delta t)} + \mathbf{F}_{\text{joint},i}^{(t+\Delta t)} \right)$$

$$\simeq \mathbf{v}_i^{(t)} + \frac{(b_\perp - b_\parallel)\Delta t}{M} \mathbf{v}_{\parallel,i}^{(t)} + \frac{\Delta t}{m} \left( \mathbf{F}_{c\kappa,i}^{(t+\Delta t)} + \mathbf{F}_{\text{joint},i}^{(t+\Delta t)} \right)$$

The approximate value of $\mathbf{v}_i^{(t+\Delta t)}$ by the above approximation is as follows.

$$\mathbf{v}_i^{(t+\Delta t)} \simeq \left[ \mathbf{v}_i^{(t)} + \frac{(b_\perp - b_\parallel)\Delta t}{M} \mathbf{v}_{\parallel,i}^{(t)} + \frac{\Delta t}{m} \left( \mathbf{F}_{c\kappa,i}^{(t+\Delta t)} + \mathbf{F}_{\text{joint},i}^{(t+\Delta t)} \right) \right] / \left( 1 + \frac{b_\perp \Delta t}{M} \right)$$

The approximate value of $\mathbf{a}_i^{(t+\Delta t)}\Delta t$ by the approximate value of $\mathbf{v}_i^{(t+\Delta t)}$ is as follows.

$$
\begin{aligned}
\mathbf{a}_i^{(t+\Delta t)}\Delta t &= \mathbf{v}_i^{(t+\Delta t)} - \mathbf{v}_i^{(t)} \\
&\simeq \left[ \mathbf{v}_i^{(t)} + \frac{(b_\perp - b_\parallel)\Delta t}{M} \mathbf{v}_{\parallel,i}^{(t)} + \frac{\Delta t}{m} \left( \mathbf{F}_{c\kappa,i}^{(t+\Delta t)} + \mathbf{F}_{\text{joint},i}^{(t+\Delta t)} \right) \right] / \left( 1 + \frac{b_\perp \Delta t}{M} \right) - \mathbf{v}_i^{(t)} \\
&= \left[ -\frac{b_\perp \Delta t}{M} \mathbf{v}_i^{(t)} + \frac{(b_\perp - b_\parallel)\Delta t}{M} \mathbf{v}_{\parallel,i}^{(t)} + \frac{\Delta t}{m} \left( \mathbf{F}_{c\kappa,i}^{(t+\Delta t)} + \mathbf{F}_{\text{joint},i}^{(t+\Delta t)} \right) \right] / \left( 1 + \frac{b_\perp \Delta t}{M} \right) \\
&= \frac{\Delta t}{m} \left[ -\frac{b_\perp}{n} \mathbf{v}_i^{(t)} + \frac{(b_\perp - b_\parallel)}{n} \mathbf{v}_{\parallel,i}^{(t)} + \mathbf{F}_{c\kappa,i}^{(t+\Delta t)} + \mathbf{F}_{\text{joint},i}^{(t+\Delta t)} \right] / \left( 1 + \frac{b_\perp \Delta t}{M} \right) \\
&= \frac{\Delta t}{m} \left[ -\frac{b_\perp}{n} (\mathbf{v}_i^{(t)} - \mathbf{v}_{\parallel,i}^{(t)}) - \frac{b_\parallel}{n} \mathbf{v}_{\parallel,i}^{(t)} + \mathbf{F}_{c\kappa,i}^{(t+\Delta t)} + \mathbf{F}_{\text{joint},i}^{(t+\Delta t)} \right] / \left( 1 + \frac{b_\perp \Delta t}{M} \right) \\
&= \frac{\Delta t}{m} \left[ -\frac{b_\perp}{n} \mathbf{v}_{\perp,i}^{(t)} - \frac{b_\parallel}{n} \mathbf{v}_{\parallel,i}^{(t)} + \mathbf{F}_{c\kappa,i}^{(t+\Delta t)} + \mathbf{F}_{\text{joint},i}^{(t+\Delta t)} \right] / \left( 1 + \frac{b_\perp \Delta t}{M} \right) \\
&= \frac{\Delta t}{m} \left[ \mathbf{F}_{b,i}^{(t)} + \mathbf{F}_{c\kappa,i}^{(t+\Delta t)} + \mathbf{F}_{\text{joint},i}^{(t+\Delta t)} \right] / \left( 1 + \frac{b_\perp \Delta t}{M} \right)
\end{aligned}
$$

If both sides are divided by $\Delta t$,

$$\mathbf{a}_i^{(t+\Delta t)} \simeq \frac{1}{\left( 1 + \frac{b_\perp \Delta t}{M} \right)} \frac{\mathbf{F}_{b,i}^{(t)} + \mathbf{F}_{c\kappa,i}^{(t+\Delta t)} + \mathbf{F}_{\text{joint},i}^{(t+\Delta t)}}{m}$$

$\mathbf{F}_{c\kappa,i}^{(t+\Delta t)}$, $\mathbf{F}_{\text{joint},i}^{(t+\Delta t)}$ satisfy the following because they are the internal forces of the worm ($\vec{0}$ is a zero vector).

$$\sum_i \mathbf{F}_{c\kappa,i}^{(t+\Delta t)} = \vec{0}$$

$$\sum_i \mathbf{F}_{\text{joint},i}^{(t+\Delta t)} = \vec{0}$$

Therefore, the approximation of the force $M\mathbf{a}_c^{(t+\Delta t)}$ received by the worm is as follows.

$$M\mathbf{a}_c^{(t+\Delta t)} = \sum_i m\mathbf{a}_i^{(t+\Delta t)}$$

$$\simeq \sum_i m \frac{1}{\left(1+\dfrac{b_\perp \Delta t}{M}\right)} \frac{\mathbf{F}_{b,i}^{(t)} + \mathbf{F}_{c\kappa,i}^{(t+\Delta t)} + \mathbf{F}_{\text{joint},i}^{(t+\Delta t)}}{m}$$

$$= \frac{1}{\left(1+\dfrac{b_\perp \Delta t}{M}\right)} \left(\sum_i \mathbf{F}_{b,i}^{(t)} + \sum_i \mathbf{F}_{c\kappa,i}^{(t+\Delta t)} + \sum_i \mathbf{F}_{\text{joint},i}^{(t+\Delta t)}\right)$$

$$= \frac{1}{\left(1+\dfrac{b_\perp \Delta t}{M}\right)} \sum_i \mathbf{F}_{b,i}^{(t)}$$

If both sides are divided by $M$,

$$\mathbf{a}_c^{(t+\Delta t)} \simeq \frac{1}{\left(1+\dfrac{b_\perp \Delta t}{M}\right)} \frac{\sum_i \mathbf{F}_{b,i}^{(t)}}{M}$$

This approximation ensures computational stability regardless of the size of $b_\perp$, $b_\parallel$. That is, this approximation solves the problem of the decrease in computational stability of numerical integration through the explicit Euler method when $b_\perp$, $b_\parallel$ are sufficiently large compared to $M/\Delta t$.

## Numerical integration of the rotational motion of i-rod using semi-implicit Euler method

First, the numerical integration formulas for $\omega_i$ and $\alpha_i$ using the implicit Euler method are as follows.

$$s_i^{(t+\Delta t)} = s_i^{(t)} + \omega_i^{(t+\Delta t)}\Delta t$$

$$\omega_i^{(t+\Delta t)} = \omega_i^{(t)} + \alpha_i^{(t+\Delta t)}\Delta t$$

If we set $\beta \equiv \frac{1}{3}\frac{b_\perp}{n}r^2$, the equation describing the rotation of i-rod is as follows.

$$I\alpha_i = \tau_{b,i} + \tau_{c\kappa,i} + \tau_{\text{joint},i}$$

$$= -\beta\omega_i + c(\omega_{i+1} - 2\omega_i + \omega_{i-1}) + \kappa(\theta_i - \theta_{i-1} - \theta_{\text{ctrl},i} + \theta_{\text{ctrl},i-1}) + \tau_{\text{joint},i}$$

$$= -\beta\omega_i + c(\omega_{i+1} - 2\omega_i + \omega_{i-1}) + \kappa(s_{i+1} - 2s_i + s_{i-1} - \theta_{\text{ctrl},i} + \theta_{\text{ctrl},i-1}) + \tau_{\text{joint},i}$$

$$= c\omega_{i-1} - (\beta + 2c)\omega_i + c\omega_{i+1} + \kappa(s_{i-1} - 2s_i + s_{i+1}) - \kappa(\theta_{\text{ctrl},i} - \theta_{\text{ctrl},i+1}) + \tau_{\text{joint},i}$$

If the above formula is expanded for time $t + \Delta t$,

$$I\alpha_i^{(t+\Delta t)} = c\omega_{i-1}^{(t+\Delta t)} - (\beta + 2c)\omega_i^{(t+\Delta t)} + c\omega_{i+1}^{(t+\Delta t)}$$
$$+ \kappa(s_{i-1}^{(t+\Delta t)} - 2s_i^{(t+\Delta t)} + s_{i+1}^{(t+\Delta t)}) - \kappa(\theta_{\text{ctrl},i}^{(t+\Delta t)} - \theta_{\text{ctrl},i+1}^{(t+\Delta t)}) + \tau_{\text{joint},i}^{(t+\Delta t)}$$

$$= c\omega_{i-1}^{(t+\Delta t)} - (\beta + 2c)\omega_i^{(t+\Delta t)} + c\omega_{i+1}^{(t+\Delta t)}$$
$$+ \kappa\left((s_{i-1}^{(t)} + \omega_{i-1}^{(t+\Delta t)}\Delta t) - 2(s_i^{(t)} + \omega_i^{(t+\Delta t)}\Delta t) + (s_{i+1}^{(t)} + \omega_{i+1}^{(t+\Delta t)}\Delta t)\right)$$
$$- \kappa(\theta_{\text{ctrl},i}^{(t+\Delta t)} - \theta_{\text{ctrl},i+1}^{(t+\Delta t)}) + \tau_{\text{joint},i}^{(t+\Delta t)}$$

$$= (c + \kappa\Delta t)\omega_{i-1}^{(t+\Delta t)} - (\beta + 2c + 2\kappa\Delta t)\omega_i^{(t+\Delta t)} + (c + \kappa\Delta t)\omega_{i+1}^{(t+\Delta t)}$$
$$+ \kappa(s_{i-1}^{(t)} - 2s_i^{(t)} + s_{i+1}^{(t)}) - \kappa(\theta_{\text{ctrl},i}^{(t+\Delta t)} - \theta_{\text{ctrl},i+1}^{(t+\Delta t)}) + \tau_{\text{joint},i}^{(t+\Delta t)}$$

The above formula is impossible to integrate because $\theta_{\text{ctrl},i}^{(t+\Delta t)}$, $\theta_{\text{ctrl},i+1}^{(t+\Delta t)}$, $\tau_{\text{joint},i}^{(t+\Delta t)}$ are unknown at time t.

$$\frac{\left|-\kappa\left((\theta_{\text{ctrl},i}^{(t+\Delta t)} - \theta_{\text{ctrl},i}^{(t)}) - (\theta_{\text{ctrl},i+1}^{(t+\Delta t)} - \theta_{\text{ctrl},i+1}^{(t)})\right) + (\tau_{\text{joint},i}^{(t+\Delta t)} - \tau_{\text{joint},i}^{(t)})\right|}{\left|(c + \kappa\Delta t)\omega_{i-1}^{(t+\Delta t)} - (\beta + 2c + 2\kappa\Delta t)\omega_i^{(t+\Delta t)} + (c + \kappa\Delta t)\omega_{i+1}^{(t+\Delta t)} + \kappa(s_{i-1}^{(t)} - 2s_i^{(t)} + s_{i+1}^{(t)}) - \kappa(\theta_{\text{ctrl},i}^{(t+\Delta t)} - \theta_{\text{ctrl},i+1}^{(t+\Delta t)}) + \tau_{\text{joint},i}^{(t+\Delta t)}\right|} \simeq 0$$

If the above formula is assumed to be true, the following approximation can be used.

$$
\begin{aligned}
I\alpha_i^{(t+\Delta t)} =\;& (c + \kappa\Delta t)\omega_{i-1}^{(t+\Delta t)} - (\beta + 2c + 2\kappa\Delta t)\omega_i^{(t+\Delta t)} + (c + \kappa\Delta t)\omega_{i+1}^{(t+\Delta t)} \\
& + \kappa(s_{i-1}^{(t)} - 2s_i^{(t)} + s_{i+1}^{(t)}) - \kappa(\theta_{\text{ctrl},i}^{(t+\Delta t)} - \theta_{\text{ctrl},i+1}^{(t+\Delta t)}) + \tau_{\text{joint},i}^{(t+\Delta t)} \\
\simeq\;& (c + \kappa\Delta t)\omega_{i-1}^{(t+\Delta t)} - (\beta + 2c + 2\kappa\Delta t)\omega_i^{(t+\Delta t)} + (c + \kappa\Delta t)\omega_{i+1}^{(t+\Delta t)} \\
& + \kappa(s_{i-1}^{(t)} - 2s_i^{(t)} + s_{i+1}^{(t)}) - \kappa(\theta_{\text{ctrl},i}^{(t)} - \theta_{\text{ctrl},i+1}^{(t)}) + \tau_{\text{joint},i}^{(t)} \\
=\;& (c + \kappa\Delta t)\omega_{i-1}^{(t+\Delta t)} - (\beta + 2c + 2\kappa\Delta t)\omega_i^{(t+\Delta t)} + (c + \kappa\Delta t)\omega_{i+1}^{(t+\Delta t)} + \tau_{\kappa,i}^{(t)} + \tau_{\text{joint},i}^{(t)}
\end{aligned}
$$

$$
\begin{aligned}
\omega_i^{(t+\Delta t)} =\;& \omega_i^{(t)} + \alpha_i^{(t+\Delta t)}\Delta t \\
=\;& \omega_i^{(t)} + \frac{\Delta t}{I}\Big( (c + \kappa\Delta t)\omega_{i-1}^{(t+\Delta t)} - (\beta + 2c + 2\kappa\Delta t)\omega_i^{(t+\Delta t)} + (c + \kappa\Delta t)\omega_{i+1}^{(t+\Delta t)} + \tau_{\kappa,i}^{(t)} + \tau_{\text{joint},i}^{(t)} \Big)
\end{aligned}
$$

The above formula can be expressed as a matrix formula as follows.

$$
\begin{bmatrix} \vdots \\ \omega_{i-1}^{(t+\Delta t)} \\ \omega_i^{(t+\Delta t)} \\ \omega_{i+1}^{(t+\Delta t)} \\ \vdots \end{bmatrix} \simeq \begin{bmatrix} \vdots \\ \omega_{i-1}^{(t)} \\ \omega_i^{(t)} \\ \omega_{i+1}^{(t)} \\ \vdots \end{bmatrix} + \frac{\Delta t}{I} \begin{bmatrix} \ddots & \vdots & \vdots & \vdots & \ddots \\ \cdots & (c+\kappa\Delta t) & -(\beta+2c+2\kappa\Delta t) & (c+\kappa\Delta t) & \cdots \\ \ddots & \vdots & \vdots & \vdots & \ddots \end{bmatrix} \begin{bmatrix} \vdots \\ \omega_{i-1}^{(t+\Delta t)} \\ \omega_i^{(t+\Delta t)} \\ \omega_{i+1}^{(t+\Delta t)} \\ \vdots \end{bmatrix} + \frac{\Delta t}{I} \begin{bmatrix} \vdots \\ \tau_{\kappa,i-1}^{(t)} + \tau_{\text{joint},i-1}^{(t)} \\ \tau_{\kappa,i}^{(t)} + \tau_{\text{joint},i}^{(t)} \\ \tau_{\kappa,i+1}^{(t)} + \tau_{\text{joint},i+1}^{(t)} \\ \vdots \end{bmatrix}
$$

In the above vector matrix formula, let us represent the vectors and matrix by the following symbols.

$$
\vec{\omega}^{(t)} \equiv \begin{bmatrix} \vdots \\ \omega_{i-1}^{(t)} \\ \omega_i^{(t)} \\ \omega_{i+1}^{(t)} \\ \vdots \end{bmatrix}
$$

$$
\mathbf{P}_{n \times n} \equiv \begin{bmatrix} \ddots & \vdots & \vdots & \vdots & \ddots \\ \cdots & (c+\kappa\Delta t) & -(\beta+2c+2\kappa\Delta t) & (c+\kappa\Delta t) & \cdots \\ \ddots & \vdots & \vdots & \vdots & \ddots \end{bmatrix}
$$

$$
\vec{\tau}_{\text{rem}}^{(t)} \equiv \begin{bmatrix} \vdots \\ \tau_{\kappa,i-1}^{(t)} + \tau_{\text{joint},i-1}^{(t)} \\ \tau_{\kappa,i}^{(t)} + \tau_{\text{joint},i}^{(t)} \\ \tau_{\kappa,i+1}^{(t)} + \tau_{\text{joint},i+1}^{(t)} \\ \vdots \end{bmatrix}
$$

Then, the matrix formula is expressed as follows.

$$
\vec{\omega}^{(t+\Delta t)} \simeq \vec{\omega}^{(t)} + \frac{\Delta t}{I}\mathbf{P}_{n \times n}\vec{\omega}^{(t+\Delta t)} + \frac{\Delta t}{I}\vec{\tau}_{\text{rem}}^{(t)}
$$

Now, the following approximation can be obtained where $\mathbf{I}_{n \times n}$ is a unit matrix of size $n \times n$.

$$
\therefore \vec{\omega}^{(t+\Delta t)} \simeq \left( \mathbf{I}_{n \times n} - \frac{\Delta t}{I}\mathbf{P}_{n \times n} \right)^{-1} \left( \vec{\omega}^{(t)} + \frac{\Delta t}{I}\vec{\tau}_{\text{rem}}^{(t)} \right)
$$

## Correction formula for the rotational inertia of the entire worm

For a floor surface with low friction like water, when numerically integrating the rotational motion of the worm, if $\Delta t > 1 \times 10^{-6}$ sec, the calculation error accumulated for the rotational inertia of the whole worm significantly influenced the calculation result $\omega_i$ and $s_i$. To prevent this, the error is corrected as follows. If $\bar{x}_i \equiv x_i - x_c$, $\bar{y}_i \equiv y_i - y_c$, the moment of inertia of the entire worm at time t is as follows by the parallel axis theorem.

$$I_{\text{body}}^{(t)} = m \sum_i \left( \left( \bar{x}_i^{(t)} \right)^2 + \left( \bar{y}_i^{(t)} \right)^2 \right) + nI$$

If i-rod is approximated as a point particle, the torque applied to the entire worm at time t is as follows where subscription x, y indicates x, y components of the vector. (See 'Numerical integration for translational motion' in Appendix)

$$\tau_{\text{body}}^{(t+\Delta t)} \simeq \frac{\sum_i \left( \bar{x}_i^{(t)} F_{b,i,\text{y}}^{(t)} - \bar{y}_i^{(t)} F_{b,i,\text{x}}^{(t)} \right)}{1 + \dfrac{b_\perp \Delta t}{M}}$$

If i-rod is approximated as a point particle, the rotational inertia of the whole worm at time $t$ is as follows.

$$L_{\text{body}}^{(t)} \simeq m \sum_i \left( \bar{x}_i^{(t)} v_{i,\text{y}}^{(t)} - \bar{y}_i^{(t)} v_{i,\text{x}}^{(t)} \right)$$

The predicted value of $\omega_i$ at $t + \Delta t$, $\omega_i^{\text{p}}$, is calculated by the semi-implicit Euler method (See 'Numerical integration of the rotational motion' in Appendix). ($\text{P} = \begin{bmatrix} \cdots & \omega_i^{\text{p}} & \cdots \end{bmatrix}^{\text{T}}$)

$$\vec{\omega}^{\text{p}} \simeq \left( \mathbf{I}_{n \times n} - \frac{\Delta t}{I} \mathbf{P}_{n \times n} \right)^{-1} \left( \vec{\omega}^{(t)} + \frac{\Delta t}{I} \vec{\tau}_{\text{rem}}^{(t)} \right)$$

The predicted value of $s_i$ at $t + \Delta t$ is as follows.

$$s_i^{\text{p}} = s_i^{(t)} + \omega_i^{\text{p}} \Delta t$$

Predicted values $x_i^{\text{p}}$, $y_i^{\text{p}}$, $\mathbf{v}_i^{\text{p}}$ for $x_i^{(t+\Delta t)}$, $y_i^{(t+\Delta t)}$, $\mathbf{v}_i^{(t+\Delta t)}$ are calculated from $\mathbf{d}_c$, $\mathbf{v}_c$, $s_i^{\text{p}}$, $\omega_i^{\text{p}}$ (See 'Minimum information required to describe the motion of each rod' in Appendix).

Using $x_i^{\text{p}}$, $y_i^{\text{p}}$, $\mathbf{v}_i^{\text{p}}$, the moment of inertia $I_{\text{body}}^{\text{p}}$ and rotational inertia $L_{\text{body}}^{\text{p}}$ of the entire worm at time $t + \Delta t$ are calculated as follows. (where $\bar{x}_i^{\text{p}} \equiv x_i^{\text{p}} - x_c^{\text{p}}$, $\bar{y}_i^{\text{p}} \equiv y_i^{\text{p}} - y_c^{\text{p}}$, $\mathbf{v}_i = \begin{bmatrix} v_{i,\text{x}} & v_{i,\text{y}} \end{bmatrix}^{\text{T}}$)

$$I_{\text{body}}^{\text{p}} = m \sum_i \left( \left( \bar{x}_i^{\text{p}} \right)^2 + \left( \bar{y}_i^{\text{p}} \right)^2 \right) + nI$$

$$L_{\text{body}}^{\text{p}} = m \sum_i \left( \bar{x}_i^{\text{p}} v_{i,\text{y}}^{\text{p}} - \bar{y}_i^{\text{p}} v_{i,\text{x}}^{\text{p}} \right)$$

$\omega_i^{(t+\Delta t)}$ is calculated as follows.

$$\omega_i^{(t+\Delta t)} = \omega_i^{\text{p}} + \frac{-\left( L_{\text{body}}^{\text{p}} - L_{\text{body}}^{(t)} \right) + \tau_{\text{body}}^{(t+\Delta t)} \Delta t}{\dfrac{I_{\text{body}}^{\text{p}} + I_{\text{body}}^{(t)}}{2}}$$

$s_i^{(t+\Delta t)}$ is calculated as follows.

$$s_i^{(t+\Delta t)} = s_i^{(t)} + \omega_i^{(t+\Delta t)} \Delta t$$

By correcting the rotational inertia for the whole worm, numerical integration of the rotational motion of the worm was well calculated even for cases when $\Delta t > 1 \times 10^{-6}$ sec, as if $\Delta t \leq 1 \times 10^{-6}$ sec.

## Proper selection of friction coefficients

When using the vertical and horizontal friction coefficients $b_{\mathrm{agar}, \perp}$, $b_{\mathrm{agar}, \parallel}$ on agar, as proposed in the previous work (*Boyle et al., 2012*), the trajectory of the escaping behavior was not accurately replicated. Therefore, we sought appropriate friction coefficients necessary for replicating the escaping behavior. We calculated new vertical and horizontal friction coefficients $b_{\eta, \perp} = \eta b_{\mathrm{agar}, \perp}$, $b_{\eta, \parallel} = \eta b_{\mathrm{agar}, \parallel}$ by multiplying scaling factor $\eta$ to the agar friction coefficients $b_{\mathrm{agar}, \perp}$, $b_{\mathrm{agar}, \parallel}$ of the previous work (*Boyle et al., 2012*). Let us denote the set of a quantity for all pairs $(i, t)$ of index $i$ and time $t$ as $\{*\}_{i,t}$. When the kymogram input $\left\{\theta_{\mathrm{ctrl},i}^{(t)}\right\}_{i,t}$ was same as *Figure 3A*, we observed how the trajectory of the escaping behavior changes with $\eta$ (*Appendix 1—figure 3A*) and analyzed representative values for each trajectory as follows (*Appendix 1—figure 3B*) (Note that when $\eta$ was less than or equal to $10^{-6}$, the time-step ($\Delta t$) was set to $10^{-6}$ sec for higher accuracy of simulation). Let us denote the average of a quantity for all pairs $(i, t)$ as $\langle * \rangle_{i,t}$. When $\eta$ was $1 \sim 10^{-9}$, the smaller $\eta$, the smaller $E_\theta = \left\langle \left| \theta_i^{(t)} - \theta_{\mathrm{ctrl},i}^{(t)} \right| \right\rangle_{i,t}$ was. The reduction in $E_\theta$ when $\eta$ changed from $10^{-2}$ to $10^{-9}$ was about 3% of the reduction when $\eta$ changed from 1 to $10^{-9}$. When $\eta$ was between 1 and $10^{-9}$, even if $\eta$ decreased, $E_\theta$ did not fall below 0.125 (rad). This is because there was a time delay between the input $\theta_{\mathrm{ctrl},i}^{(t)}$ and the response $\theta_i^{(t)}$. As $\eta$ decreased from 1 to $10^{-2}$, the total traveled distance of the worm ($\sum_t \left| \mathbf{v}_c^{(t)} \Delta t \right|$) and the total absolute angle change ($\mathcal{S} = \sum_{t=0}^{T-\Delta t} \left| \left\langle s_i^{(t+\Delta t)} \right\rangle_i - \left\langle s_i^{(t)} \right\rangle_i \right|$) increased, and the trajectory became more similar to the experimental video (*Broekmans et al., 2016*). When $\eta$ was between $10^{-2}$ and $10^{-6}$, the worm's trajectory was almost identical, and thus the total traveled distance, $\mathcal{S}$, and the pattern of $\left| \mathbf{F}_{b,i}^{(t)} \right|$ (*Appendix 1—figure 4*) were similar across trajectories. If $\eta$ was smaller than $10^{-6}$, the worm's total traveled distance decreased and $\mathcal{S}$ increased, and the trajectory was no longer similar to the experimental video. This was because a too small friction coefficients hindered the worm from obtaining enough propulsive force from the ground (*Appendix 1—figure 4*).

The same method with the kymogram input $\left\{\theta_{\mathrm{ctrl},i}^{(t)}\right\}_{i,t}$ same as *Figure 3B* used to analyze the effect of $\eta$ on the trajectory of the escaping behavior was applied to analyze the trajectory of the delta-turn according to $\eta$ (*Appendix 1—figure 5A*). When $\eta$ was $1 \sim 10^{-9}$, the smaller $\eta$, the smaller $E_\theta$ was (*Appendix 1—figure 5B*). The reduction in $E_\theta$ when $\eta$ changed from $10^{-2}$ to $10^{-9}$ was about 3% of the reduction when $\eta$ changed from 1 to $10^{-9}$. When $\eta$ was between 1 and $10^{-9}$, even if $\eta$ decreased, $E_\theta$ did not fall below 0.15 (rad). As $\eta$ decreased from 1 to $10^{-2}$, $\mathcal{S}$ increased, and the trajectory became more similar to the experimental video. When $\eta$ was between $10^{-2}$ and $10^{-6}$, the worm's trajectory was almost identical, and thus the total traveled distance, $\mathcal{S}$, and the pattern of $\left| \mathbf{F}_{b,i}^{(t)} \right|$ (*Appendix 1—figure 6*) were similar across trajectories. If $\eta$ was smaller than $10^{-6}$, the worm's total traveled distance decreased and $\mathcal{S}$ increased, and the trajectory was no longer similar to the experimental video. This is because a too small friction coefficients hindered the worm from obtaining enough propulsive force from the ground (*Appendix 1—figure 6*).

In conclusion, when the ratio between vertical and horizontal friction coefficients was constant at $40 (= b_{\eta, \perp}/b_{\eta, \parallel} = b_{\mathrm{agar}, \perp}/b_{\mathrm{agar}, \parallel})$, selecting an appropriate $\eta$ value (between $10^{-2}$ and $10^{-6}$) was crucial for replicating the trajectory of sequenced locomotive behavior. Therefore, we chose $b_{\eta, \perp}$, $b_{\eta, \parallel}$ ($\eta = 10^{-2}$) as the friction coefficients that sufficiently reduce $E_\theta$ among those closest to the agar friction coefficients of the previous work (*Boyle et al., 2012*) for both escaping behavior and delta-turn.

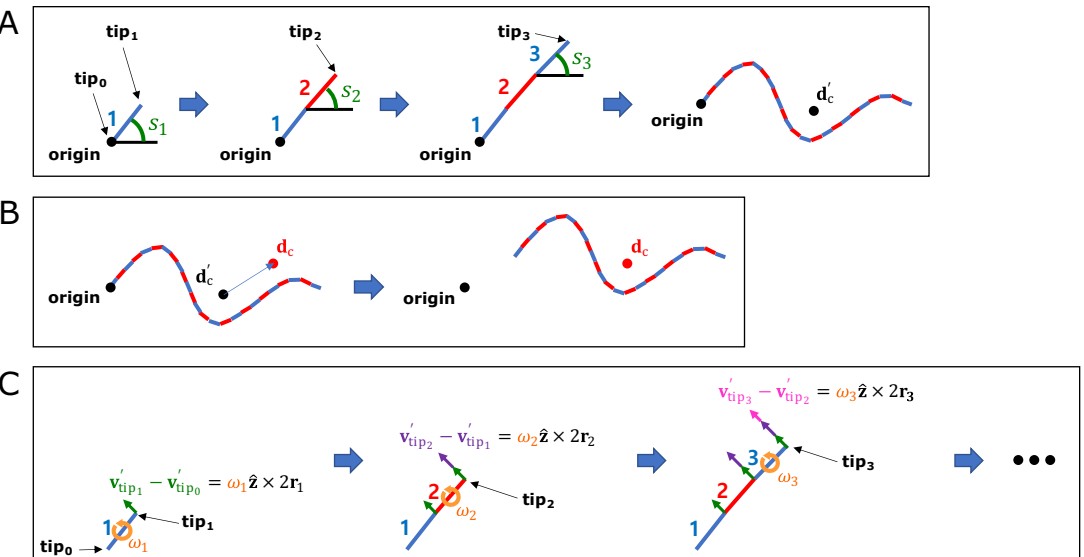

**Appendix 1—figure 1.** Method of compressing motion state information. (**A**) Method of calculating the relative position of the rod. (**B**) Method of calculating the absolute position of the rod. (**C**) Method of calculating the relative velocity of the rod.

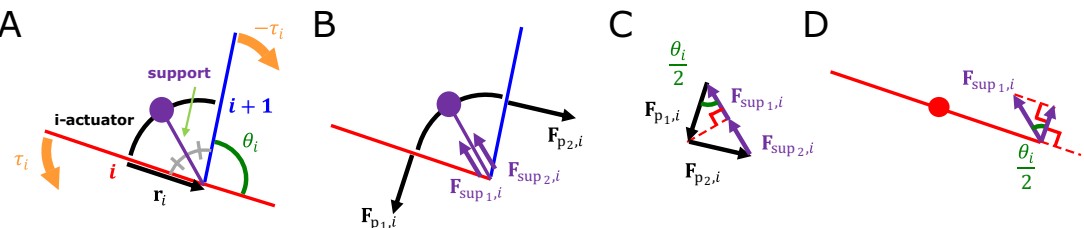

**Appendix 1—figure 2.** The i-actuator. (**A**) Composition of i-actuator. (**B**) Forces that i-actuator gives to i-rod and (i+1)-rod. (**C**) Resultant force given by i-actuator. (**D**) Force component of i-actuator that applies torque to i-rod.

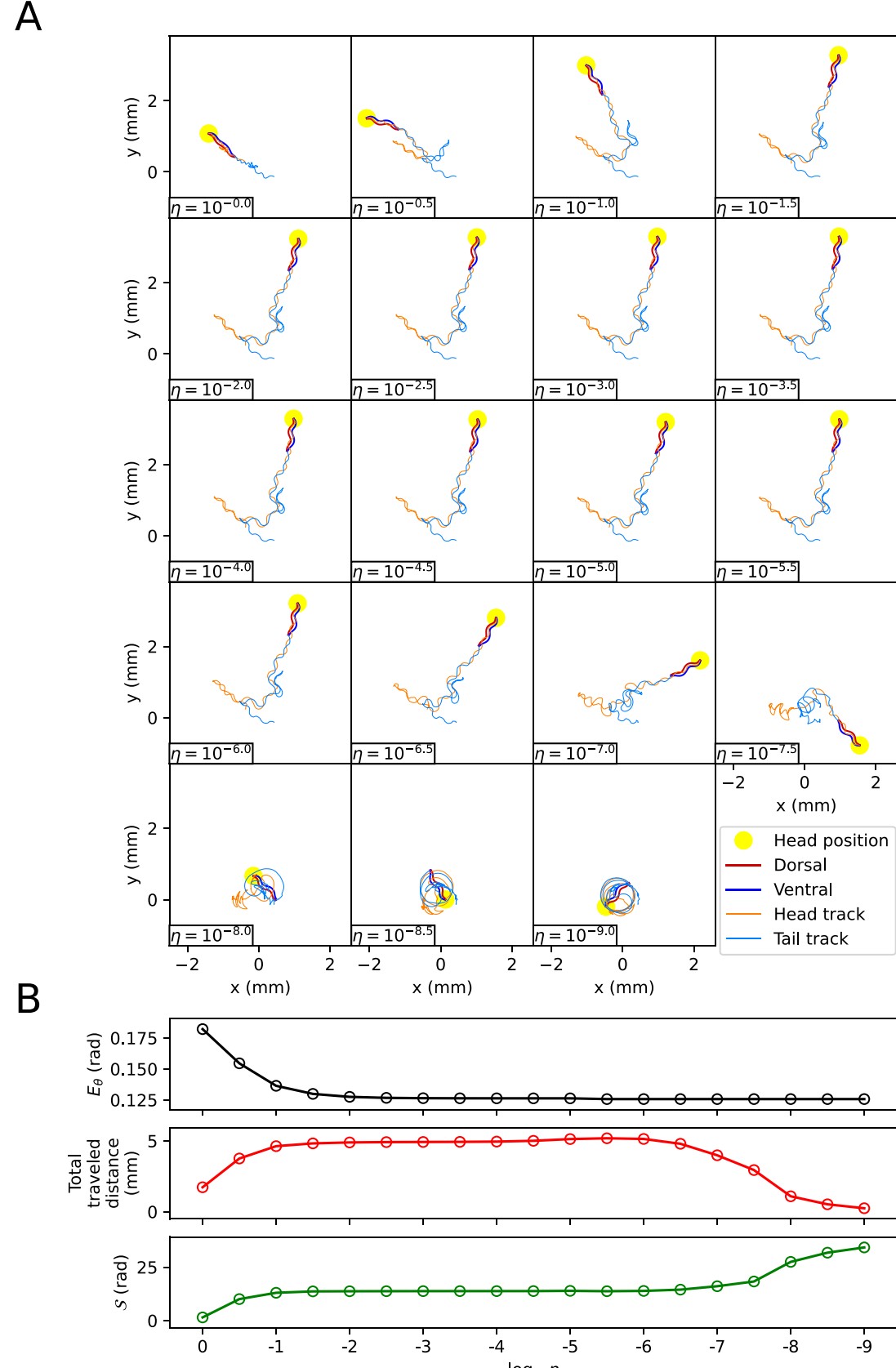

**Appendix 1—figure 3.** The effect of the scaling factor $\eta$ of the friction coefficients on the escaping behavior. (**A**) *Appendix 1—figure 3 continued on next page*

*Appendix 1—figure 3 continued*

The trajectory of the worm for each scaling factor $\eta$. (**B**) Characteristics of the trajectory. The top graph represents $E_\theta = \left\langle \left| \theta_i^{(t)} - \theta_{\text{ctrl},i}^{(t)} \right| \right\rangle_{i,t}$. The middle graph shows the total traveled distance of the worm. The bottom graph represents the total absolute angle change ($\mathcal{S} = \sum_{t=0}^{T-\Delta t} \left| \left\langle s_i^{(t+\Delta t)} \right\rangle_i - \left\langle s_i^{(t)} \right\rangle_i \right|$, where $T$ is the total time of the experimental video).

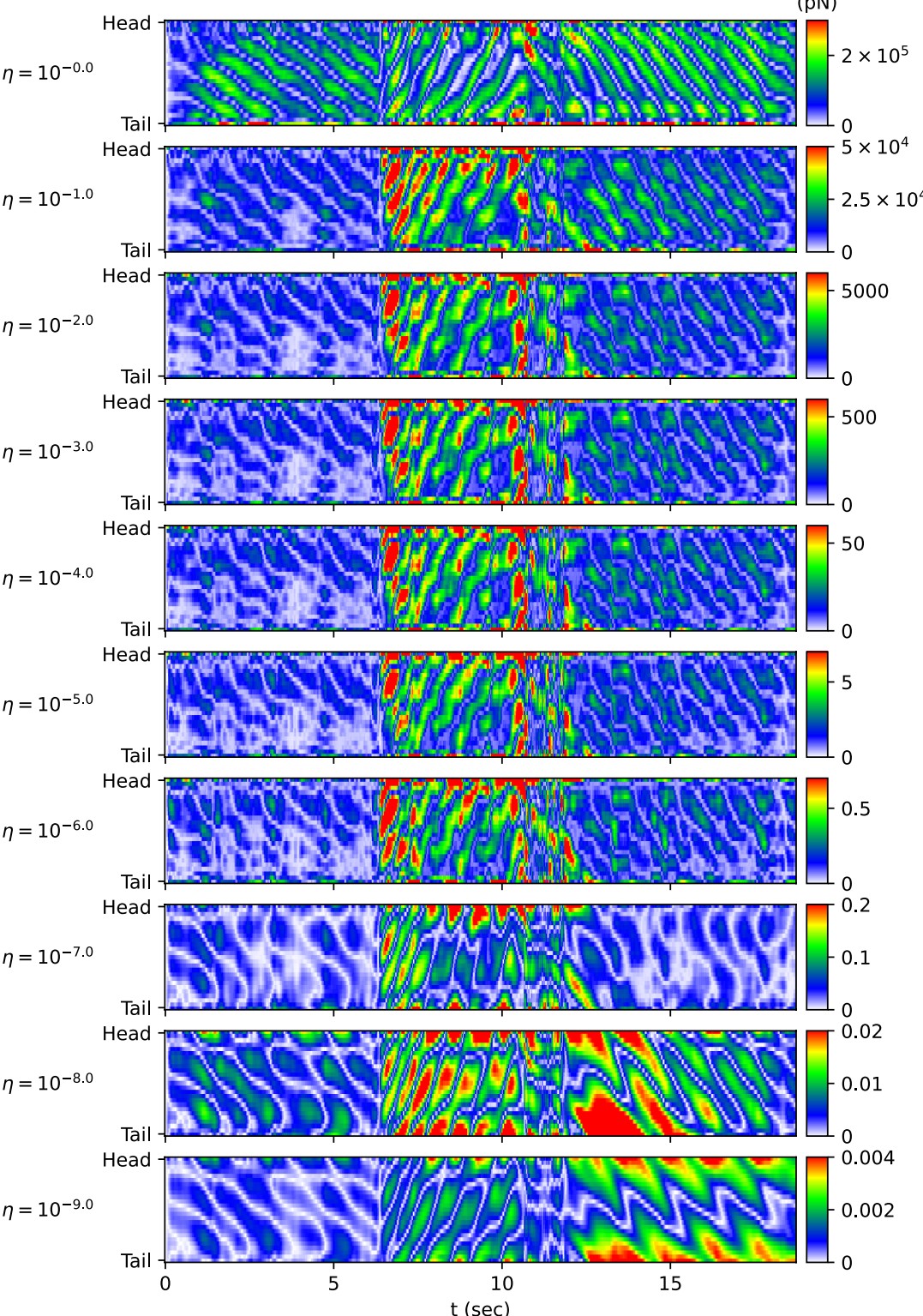

**Appendix 1—figure 4.** The magnitude of the frictional force $\left|\mathbf{F}_{b,i}^{(t)}\right|$ during the escaping behavior depending on the scaling factor $\eta$ of the friction coefficient. The color of each point in the heatmap represents the value of $\left|\mathbf{F}_{b,i}^{(t)}\right|$.

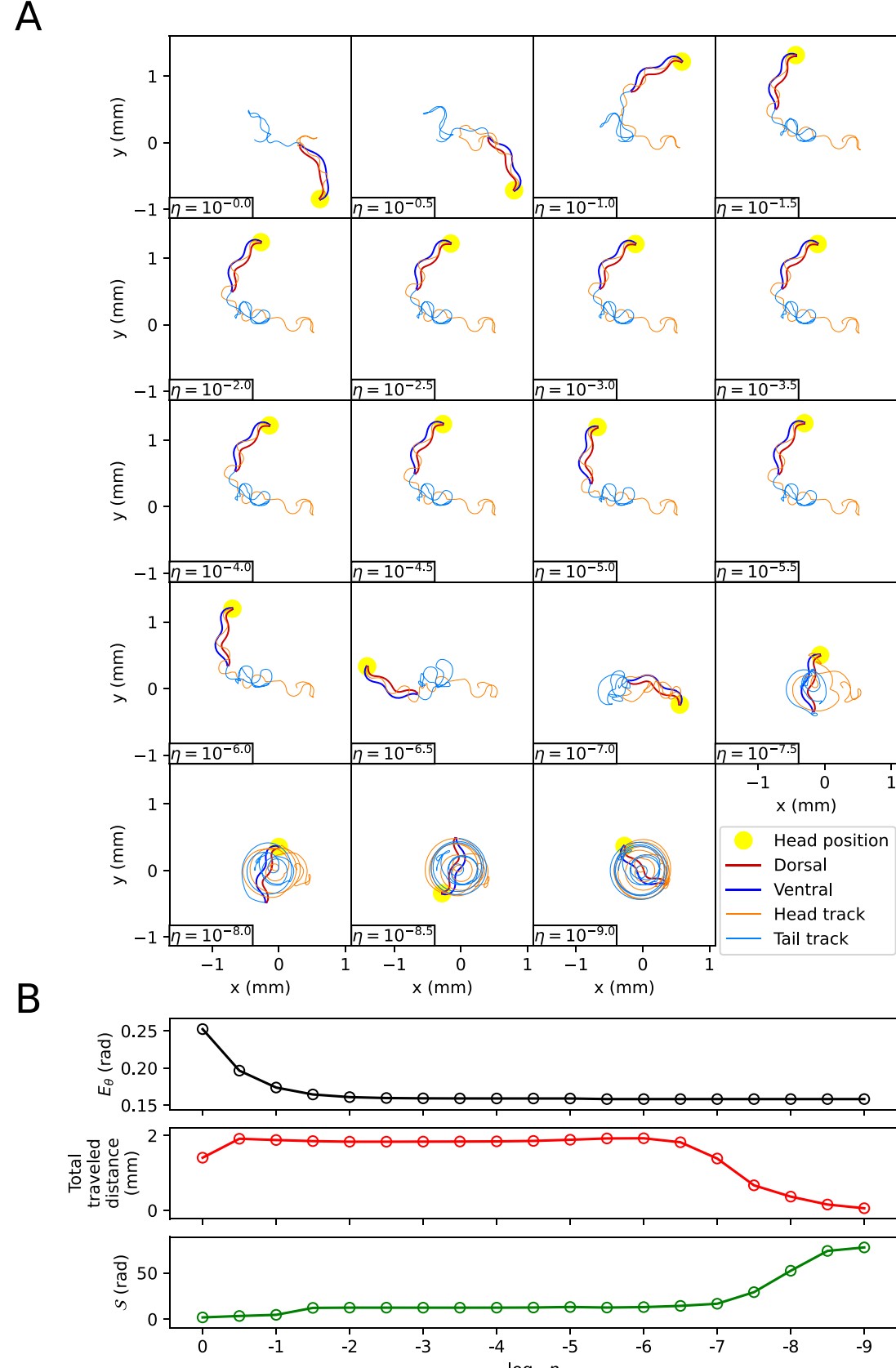

**Appendix 1—figure 5.** The impact of the scaling factor $\eta$ of the friction coefficients on the delta-turn. (**A**) The
*Appendix 1—figure 5 continued on next page*

*Appendix 1—figure 5 continued*

trajectory of the worm for each scaling factor $\eta$. (**B**) Characteristics of the trajectory. The top graph represents
$E_\theta = \left\langle \left| \theta_i^{(t)} - \theta_{\text{ctrl},i}^{(t)} \right| \right\rangle_{i,t}$. The middle graph shows the total traveled distance of the worm. The bottom graph
represents the total absolute angle change ($\mathcal{S} = \sum_{t=0}^{T-\Delta t} \left| \left\langle s_i^{(t+\Delta t)} \right\rangle_i - \left\langle s_i^{(t)} \right\rangle_i \right|$)

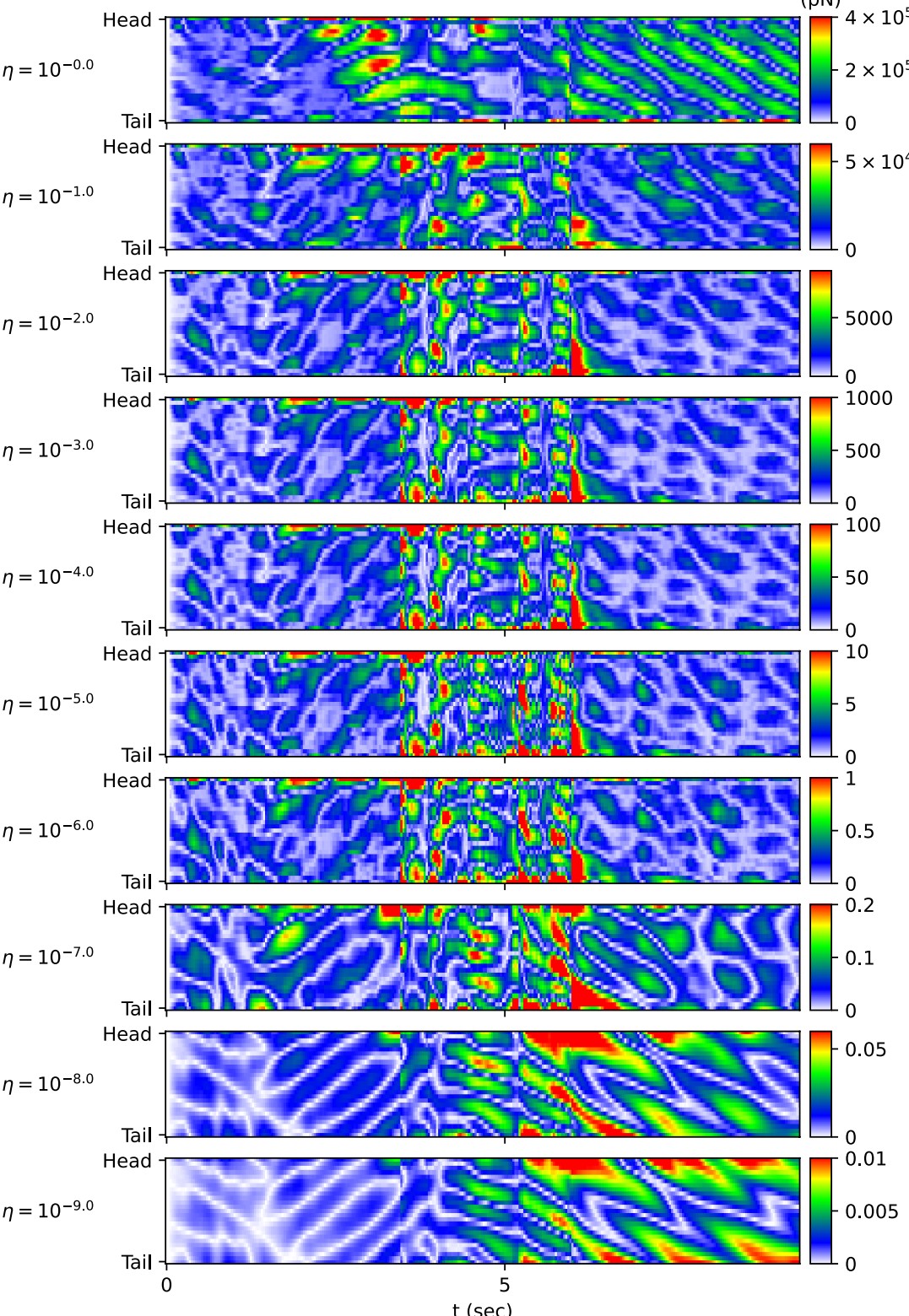

**Appendix 1—figure 6.** The magnitude of the frictional force $\left|\mathbf{F}_{b,i}^{(t)}\right|$ during the delta-turn depending on the scaling factor $\eta$ of the friction coefficient. The color of each point in the heatmap represents the value of $\left|\mathbf{F}_{b,i}^{(t)}\right|$.

