## [Editor Report · eLife assessment]

This **useful** study introduces a simple mechanical model of *C. elegans* locomotion that captures aspects of the worm's behavioral repertoire beyond forward crawling. While the kinetic model (ElegansBot) provides a compromise and starting point to help understand the mechanical components of *C. elegans* behavior, the claim that this work improves on extant mechanical models is **incomplete**, including modeling a 3-dimensional turning behavior with a 2-dimensional model without sufficient justification. In addition, the results of the application of the model to previously unstudied behaviors are primarily qualitative and do not produce new predictions.

---

## [Referee Report · Reviewer #1 (Public Review)]

Summary:

This work describes a simple mechanical model of worm locomotion, using a series of rigid segments connected by damped torsional springs and immersed in a viscous fluid. It uses this model to simulate forward crawling movement, as well as omega turns.

Strengths:

The primary strength is in applying a biomechanical model to omega-turn behaviors. The biomechanics of nematode turning behaviors are relatively less well described and understood than forward crawling, and the increase in power during omega turns is one of the more novel results. The model itself may be a useful implementation to other researchers, particularly owing to its simplicity.

Weaknesses:

The strength of the model presented in this work relative to prior approaches is not well supported, and in general the paper would be improved with a better description of the broader context of existing modeling literature related to undulatory locomotion. This paper claims to improve on previous approaches to taking body shapes as inputs. However, the sole nematode model cited aims to do something different, and arguably more significant, which is to use experimentally derived parameters to model both the neural circuits that induce locomotion as well as the biomechanics and to subsequently compare the model to experimental data. Other modeling approaches do take experimental body kinematics as inputs and use them to produce force fields, however, they are not cited or discussed. Finally, the overall novelty of the approach is questionable. A functionally similar approach was developed in 2012 to describe worm locomotion in lattices (Majmudar, 2012, Roy. Soc. Int.), which is not discussed and would provide an interesting comparison and needed context.

In some sense, because the model takes kinematics as an input and uses previously established techniques to model mechanics, it is unsurprising that it can reproduce experimentally observed kinematics, however, the forces calculated and the variation of parameters could be of interest, but other methods derived from kinematics could provide similar results. It is unclear what the predictive power of the model is.

Relatedly, a justification of why the drag coefficients had to be changed by a factor of 100 should be explored. Plate conditions are difficult to replicate and the rheology of plates likely depends on several factors, but is for example, changes in hydration level likely to produce a 100-fold change in drag? or something more interesting/subtle within the model producing the discrepancy?

Finally, the language used to distinguish different modeling approaches was often unclear. For example, it was unclear in what sense the model presented in Boyle, 2012 was a "kinetic model" and in many situations, it appeared that the term kinematic might have been more appropriate. Other phrases like "frictional forces caused by the tension of its muscles" were unclear at first glance, and might benefit from revision and more canonical usage of terms.

---

## [Referee Report · Reviewer #2 (Public Review)]

Summary:

Developing a mechanical model of *C. elegans* is difficult to do from basic principles because it moves at low (but not very small) Reynolds number, is itself visco-elastic, and often is measured moving at a solid/liquid interface. The ElegansBot is a good first step at a kinetic model that reproduces a wide range of *C. elegans* motility behavior.

Strengths:

The model is general due to its simplicity and likely useful for various undulatory movements. The model reproduces experimental movement data using realistic physical parameters (e.g. drags, forces, etc). The model is predictive (semi?) as shown in the liquid to solid gait transition. The model is straightforward in implementation and so likely is adaptable to modification and addition of control circuits.

Comments on revised version:

This is a revised manuscript. I'm happy with the changes made, including the specific responses to my previous concerns.

---

## [Referee Report · Reviewer #3 (Public Review)]

A mechanical model of *C. elegans*, embedded in a resistive force environment, is used to calculate input torque patterns required to generate output curvature patterns and coordinates, corresponding to a number of different locomotion behaviors in *C. elegans*.

Strengths:

The use of a mechanical model to study a variety of locomotor sequences and the grounding in empirical data are strengths. The matching of speeds (though requiring adjusted drag coefficients) is a strength.

Weaknesses:

The paper lacks evidence of numerical validation or comparison with the results and tools in the literature. E.g. is it surprising that the uniform torque distribution yields maximal speed? What is the relation between input and output data? How does the input-output relation depend on the parameters of the model? What novel model predictions are made?

In particular, if validated, the breakdown of drag forces and torque distributions during forward locomotion and turning behaviors may be interesting to compare to predictions by other tools, and to empirical measurement. One caveat is that the worm touches itself during such turns, and even crosses over itself in delta turns, and so the estimated drag coefficients and the resultant mechanical forces are likely incorrect.

---

## [Author Response]

The following is the authors’ response to the original reviews.

**Reviewer #1**

**Public Review**
Summary:(1) This work describes a simple mechanical model of worm locomotion, using a series of rigid segments connected by damped torsional springs and immersed in a viscous fluid.(2) It uses this model to simulate forward crawling movement, as well as omega turns.Strengths:(3) The primary strength is in applying a biomechanical model to omega-turn behaviors.(4) The biomechanics of nematode turning behaviors are relatively less well described and understood than forward crawling.(5) The model itself may be a useful implementation to other researchers, particularly owing to its simplicity.Weaknesses:(6) The strength of the model presented in this work relative to prior approaches is not well supported, and in general, the paper would be improved with a better description of the broader context of existing modeling literature related to undulatory locomotion.(7) This paper claims to improve on previous approaches to taking body shapes as inputs.(8) However, the sole nematode model cited aims to do something different, and arguably more significant, which is to use experimentally derived parameters to model both the neural circuits that induce locomotion as well as the biomechanics and to subsequently compare the model to experimental data.(9) Other modeling approaches do take experimental body kinematics as inputs and use them to produce force fields, however, they are not cited or discussed.(10) Finally, the overall novelty of the approach is questionable.(11) A functionally similar approach was developed in 2012 to describe worm locomotion in lattices (Majmudar, 2012, Roy. Soc. Int.), which is not discussed and would provide an interesting comparison and needed context.

9-11: The paper you recommended and our manuscript have some similarities and differences.

Similarities

Firstly, the components constituting the worm are similar in both models. ElegansBot models the worm as a chain of n rods, while the study by Majmudar et al. (2012) models it as a chain of n beads. Each bead in the Majmudar et al. model has a directional vector, making it very similar to ElegansBot's rod. However, there's a notable difference: in the Majmudar et al. model, each bead has an area for detecting contact between the obstacle and the bead, while in ElegansBot, the rod does not feature such an area.

Secondly, the types of forces and torques acting on the components constituting the worm are similar. Each rod in ElegansBot receives frictional force, muscle force, and joint force. Each bead in the Majmudar et al. model receives a constraint force, viscous force, and a repulsive force from obstacles. Each rod in ElegansBot receives frictional torque, muscle torque, and joint torque. Each bead in the Majmudar et al. model receives elastic torque, constraint torque, drive torque, and viscous torque. The Majmudar et al. model's constraint force and torque are similar to ElegansBot's joint force and torque in that they prevent two connected components of the worm from separating. The Majmudar et al. model's viscous force and torque are similar to ElegansBot's frictional force and torque in that they are forces exchanged between the worm and its surrounding environment (ground surface). The Majmudar et al. model's drive torque is similar to ElegansBot's muscle force and muscle torque as a cause of the worm's motion. However, unlike ElegansBot, the Majmudar et al. model did not consider the force generating the drive torque, and there are differences in how each force and torque is calculated. This will be discussed in more detail below.

Differences

Firstly, the medium in which the worm locomotes is different. ElegansBot is a model describing motion in a homogeneous medium like agar or water without obstacles, while the Majmudar et al. model describes motion in water with circular obstacles fixed at each lattice point. This is because the purposes of the models are different. ElegansBot analyzes locomotion patterns based on the friction coefficient, while the Majmudar et al. model analyzes locomotion patterns based on the characteristics of the obstacle lattice, such as the distance between obstacles. Also, for this reason, the Majmudar et al. model's bead, unlike ElegansBot's rod, receives a repulsive force from obstacles.

Secondly, the specific methods of calculating similar types of forces differ. ElegansBot calculates joint forces by substituting frictional forces, muscle forces, frictional torques, and muscle torques into an equation derived from differentiating a boundary condition equation twice over time, where two neighboring rods always meet at one point. This involves determining the process through which various forces and torques are transmitted across the worm. Specifically, it entails calculating how the frictional forces and torques, as well as the muscle forces and torques acting on each rod, are distributed throughout the entire length of the worm. In contrast, The Majmudar et al. model uses Lagrange multipliers method based on a boundary condition that the curve length determined by each bead's tangential angle does not change, to calculate the constraint force and torque before calculating the drive torque and viscous force. This implies that the Majmudar et al. model did not consider the mechanism by which the drive torque and viscous force received by one bead are distributed throughout the worm. ElegansBot's rod receives an anisotropic Stokes frictional force from the ground surface, while the Majmudar et al. model considered the frictional force according to the Navier-Stokes equation for incompressible fluid, assuming the fluid velocity at the bead's location as the bead's velocity.

Thirdly, unlike the Majmudar et al. model, ElegansBot considers the inertia of the worm components. Therefore, ElegansBot can simulate regardless of how low or high the ground surface's friction coefficient is. the Majmudar et al. model is not like this.

(12) The idea of applying biomechanical models to describe omega turns in *C. elegans* is a good one, however, the kinematic basis of the model as used in this paper (the authors do note that the control angle could be connected to a neural model, but don't do so in this work) limits the generation of neuromechanical control hypotheses.

8, 12: We do not agree with the claim that ElegansBot could limit other researchers in generating neuromechanical control hypotheses. The term θ_("ctrl" ,i)^((t) ) used in our model is designed to be replaceable with neuromechanical control in the future.

(13) The model may provide insights into the biomechanics of such behaviors, however, the results described are very minimal and are purely qualitative.(14-1) Overall, direct comparisons to the experiments are lacking or unclear.

14-1: If you look at the text explaining Fig. 2 and 5 (Fig. 2 and 4 in old version), it directly compares the velocity, wave-number, and period as numerical indicators representing the behavior of the worm, between the experiment and ElegansBot.

(14-2) Furthermore, the paper claims the value of the model is to produce the force fields from a given body shape, but the force fields from omega turns are only pictured qualitatively.

13, 14-2: We gratefully accept the point that our analysis of the omega-turn is qualitative. Therefore, we have conducted additional quantitative analysis on the omega-turn and inserted the results into the new Fig. 4. We have considered the term 'Force field' as referring to the force vector received by each rod. We have created numerical indicators representing various behaviors of the worm and included them in the revised manuscript.

(15) No comparison is made to other behaviors (the force experienced during crawling relative to turning for example might be interesting to consider) and the dependence of the behavior on the model parameters is not explored (for example, how does the omega turn change as the drag coefficients are changed).

Thank you for the great idea. To compare behaviors, first, a clear criterion for distinguishing behaviors is needed. Therefore, we have created a new mathematical definition for behavior classification in the revised manuscript (“Defining Behavioral Categories” in Method). After that, we compared the force and power (energy consuming rate) between each forward locomotion, backward locomotion, and omega-turn (Fig. 4). And in the revised manuscript, we newly analyzed how the turning behavior changes with variations in the friction coefficients in Figs. S4-S7.

(16) If the purpose of this paper is to recapitulate the swim-to-crawl transition with a simple model, and then apply the model to new behaviors, a more detailed analysis of the behavior of the model variables and their dependence on the variables would make for a stronger result.

In our revised manuscript, we have quantitatively analyzed the changes occurring in turning behavior from water to agar, and the results are presented in Figs. S9 and S10.

(17) In some sense, because the model takes kinematics as an input and uses previously established techniques to model mechanics, it is unsurprising that it can reproduce experimentally observed kinematics, however, the forces calculated and the variation of parameters could be of interest.(18) Relatedly, a justification of why the drag coefficients had to be changed by a factor of 100 should be explored.(19) Plate conditions are difficult to replicate and the rheology of plates likely depends on a number of factors, but is for example, changes in hydration level likely to produce a 100-fold change in drag? or something more interesting/subtle within the model producing the discrepancy?

18, 19: As mentioned in the paper, we do not know if the friction coefficients in the study of Boyle et al. (2012) and the friction coefficients in the experiment of Stephens et al. (2016) are the same. In our revised manuscript, we have explored more in detail the effects of the friction coefficient's scale factor, and explained why we chose a scale factor of 1/100 (“Proper Selection of Friction Coefficients” in Supplementary Information). In summary, we analyzed the changes in trajectory due to scaling of the friction coefficient, and chose the scale factor 1/100 as it allowed ElegansBot to accurately reproduce the worm's trajectory while also being close to the friction coefficients in the Boyle et al. paper.

(20) Finally, the language used to distinguish different modeling approaches was often unclear.(21) For example, it was unclear in what sense the model presented in Boyle, 2012 was a "kinetic model" and in many situations, it appeared that the term kinematic might have been more appropriate.Thank you for the feedback. As you pointed it out, we have corrected that part to 'kinematic' in the revised manuscript.(22) Other phrases like "frictional forces caused by the tension of its muscles" were unclear at first glance, and might benefit from revision and more canonical usage of terms.

We agree that the expression may not be immediately clear. This is due to the word limit for the abstract (the abstract of eLife VOR should be under 200 words, and our paper's abstract is 198 words), which forced us to convey the causality in a limited number of words. Therefore, although we will not change the abstract, the expression in question means that the muscle tension, which is the cause of the worm's locomotion, ultimately generates the frictional force between the worm and the ground surface.

**Recommendations For The Authors**
(23) As I stated in my public review, I think the paper could be made much stronger if a more detailed exploration of turning mechanics was presented.(24) Relatedly, rather than restricting the analysis to individual videos of turning behaviors, I wonder if a parameterized model of the turning kinematics would be fruitful to study, to try to understand how different turning gaits might be more or less energetically favorable.

We thank the reviewer once again for their suggestion. Thanks to their proposal, we were able to conduct additional quantitative analysis on turning behavior.

**Reviewer #2**

**Public Review**
Summary:(1) Developing a mechanical model of *C. elegans* is difficult to do from basic principles because it moves at a low (but not very small) Reynolds number, is itself visco-elastic, and often is measured moving at a solid/liquid interface.(2) The ElegansBot is a good first step at a kinetic model that reproduces a wide range of *C. elegans* motiliy behavior.Strengths:(3) The model is general due to its simplicity and likely useful for various undulatory movements.(4) The model reproduces experimental movement data using realistic physical parameters (e.g. drags, forces, etc).(5) The model is predictive (semi?) as shown in the liquid-to-solid gait transition.(6) The model is straightforward in implementation and so likely is adaptable to modification and addition of control circuits.Weaknesses:(7) Since the inputs to the model are the actual shape changes in time, parameterized as angles (or curvature), the ability of the model to reproduce a realistic facsimile of *C. elegans* motion is not really a huge surprise.(8) The authors do not include some important physical parameters in the model and should explain in the text these assumptions.(9. 1) The cuticle stiffness is significant and has been measured [1].(10. 2) The body of *C. elegans* is under high hydrostatic pressure which adds an additional stiffness [2].(11. 3) The visco-elasticity of *C. elegans* body has been measured. [3]

Thank you for asking. The stiffness of *C. elegans* is an important consideration. We took this into account when creating ElegansBot, but did not explain it in the paper. The detailed explanation is as follows. *C. elegans* indeed has stiffness due to its cuticle and internal pressure. This stiffness is treated as a passive elastic force (elastic force term of lateral passive body force) in the paper of Boyle et al. (2012). However, the maximum spring constant of the passive elastic force is 1/20 of the maximum spring constant of the active elastic force. If we consider this fact in our model, the elastic term of the muscle torque (τ_κ_) is as follows: (κ_1_ is the active torque elasticity coefficient, κ_2_ is the passive torque elasticity coefficient)τκ=κ1(θ−θctrl )+κ2(θ−0)=(κ1+κ2)(θ−κ1κ1+κ2θctrl )=κ(θ−θ~ctrl )

where

θ~ctrl ≡κ1κ1+κ2θctrl 

Therefore, there is no need to describe the active and passive terms separately in τ_κ_

Furthermore, since κ2=κ120, assuming κ≃κ1, then θ~ctrl ≃θctrl  and τκ≃κ(θ−θctrl ).

(12) There is only a very brief mention of proprioception.(13) The lack of inclusion of proprioception in the model should be mentioned and referenced in more detail in my opinion.

As you emphasized, proprioception is an important aspect in the study of *C. elegans*' locomotion. In our paper, its importance is briefly introduced with a sentence each in the introduction and discussion. However, our research is a model about the process of the creation of body motion originated from muscle forces, and it does not model the sensory system that senses body posture. Therefore, there is no mention of using proprioception in our paper's results section. What is mentioned in the discussion is that ElegansBot can be applied as the kinetic body model part in a combination model of a kinetic body model and a neuronal circuit model that receives proprioception as a sensory signal.

(14) These are just suggested references.(15) There may be more relevant ones available.

The papers you provided contain specific information about the Young's modulus of the *C. elegans* body. The first paper (Rahimi et al., 2022) measured the Young's modulus of the cuticle after chemically isolating it from *C. elegans*, while the second paper (Park et al., 2007) and third paper (Backholm et al., 2013) measured the elasticity and Young's modulus of *C. elegans* without separating the cuticle. Based on the Young's modulus provided in each paper (although the second and third papers did not measure stiffness in the longitudinal direction), we derived the elastic coefficient (assuming a worm radius of 25 μm, cuticle thickness of 0.5 μm, and 1/25 of longitudinal length of the cuticle of 40 μm). The range was quite broad, from 9.82ⅹ1011 μg/sec2 (from the first paper) to 2.16 ⅹ 108 μg / sec2 (from the third paper). Although the elastic coefficient value in our paper falls within this range, since the range of the elastic coefficient is wide, we think we can modify the elastic coefficient in our paper and will be able to reapply our model if more accurate values become known in the future.

**Reviewer #3**

**Public Review**
Summary:(1) A mechanical model is used with input force patterns to generate output curvature patterns, corresponding to a number of different locomotion behaviors in *C. elegans*Strengths:(2) The use of a mechanical model to study a variety of locomotor sequences and the grounding in empirical data are strengths.(3) The matching of speeds (though qualitative and shown only on agar) is a strength.Weaknesses:(4) What is the relation between input and output data?

ElegansBot takes the worm's body control angle as the input, and produces trajectory and force of each segment of the worm as the output.

(5) How does the input-output relation depend on the parameters of the model?

If 'parameter' is understood as vertical and horizontal friction coefficients, then the explanation for this can be found in Fig. 5 (Fig. 4 in the old version).

(6) What biological questions are addressed and can significant model predictions be made?

Equation of motion deciphering locomotion of *C. elegans* including turning behaviors which were relatively less well understood.

**Recommendations For The Authors**
(7) The novelty and significance of the paper should be clarified.

We have added quantitative analyses of turning behavior in the revised manuscript, and we hope this will be helpful to you.

(8) Previously much more detailed models have been published, as compared to this one.

We hope the reviewer can point out any previous model that we may have missed.

(9) The mechanics here are simplified (e.g. no information about dorsal/ventral innervation but only a bending angle) setting limitations on the capacity for model predictiveness.(10) Such limitations should be discussed.

We view the difference between dorsal/ventral innervation and bending angle not as a matter of simplification, but rather as a reflection of the hierarchy that our model implements. Our model does not consider dorsal/ventral innervation, but it uses the bending angle to reproduce behavior in various input and frictional environments, which signifies the strong predictiveness of ElegansBot (Figure 2, 3, 5 (2, 3, 4 in the old version)). Moreover, if the midline of *C. elegans* is incompressible, then modeling by dividing into dorsal/ventral, as opposed to modeling solely with the bending angle, does not increase the degree of freedom of the worm model, and therefore does not increase its predictiveness.

(11) The aims of the paper and results need to be supported quantitatively and analyzed through parameter sweeps and intervention.

We have conducted additional quantitative analyses on turning behavior as suggested by Reviewer #1 (Fig. 4, S4-S7, S9, and S10).

(12) The methods are given only in broad brushstrokes, and need to be much more clear (and ideally sharing all code).

We have thoroughly detailed every aspect of this research, from deriving the physical constants of *C. elegans,* agar, and water to developing the formulas and proofs necessary for operating ElegansBot and its applications. This comprehensive information is all presented in the Results, Methods, and Supplementary Information sections, as well as in the source code. Moreover, we have already ensured that our research can be easily reproduced by providing detailed explanations and by making ElegansBot accessible through public software databases (PyPI, GitHub). To further aid in its application and understanding, especially for those less familiar with the subject, we have also included minimal code as examples in the database. This code is designed to simplify the process of reproducing the results of the paper, thereby making our research more accessible and understandable. Therefore, we believe that readers will easily gain significant assistance from the extensive information we have provided. Should readers require further help, they can always contact us, and we will be readily available to offer support.

(13) The supporting figures and movies need to include a detailed analysis to evidence the claims.

We have conducted and provided additional quantitative analyses on turning behavior as suggested by Reviewer #1 (Fig. 4, S4-S7, S9, and S10).